# Exploring exogenous controls on short- versus long-term erosion rates globally

Shiuan-An Chen[1], Katerina Michaelides[1,2,3], David A. Richards[1,2], Michael Bliss Singer[3,4,5]

[1]School of Geographical Sciences, University of Bristol, Bristol, BS8 1SS, UK
[2]Cabot Institute for the Environment, University of Bristol, Bristol, UK
[3]Earth Research Institute, University of California Santa Barbara, Santa Barbara, California 91306, USA
[4]School of Earth and Environmental Sciences, Cardiff University, Cardiff, CF10 3AT, UK
[5]Water Research Institute, Cardiff University, Cardiff, CF10 3AX, UK

*Correspondence to*: Shiuan-An Chen (b95208027@gmail.com) and Katerina Michaelides (katerina.michaelides@bristol.ac.uk)

**Abstract.** Erosion is directly tied to landscape evolution through the relationship between sediment flux and vertical lowering of the land surface. Therefore, the analysis of erosion rates across the planet measured over different temporal domains may provide perspectives on the drivers and processes of land surface change over different timescales. Different metrics are commonly used to quantify erosion (or denudation) over timescales of $< 10^1$ y (suspended sediment flux) and $10^3$–$10^6$ y (cosmogenic radionuclides), so reconciling potentially contrasting rates at these timescales at any location is challenging.
Studies over the last several decades into erosion rates and their controls have yielded valuable insights into geomorphic processes and landforms over time and space, but many are focused at local/regional scales. Gaps remain in understanding large-scale patterns and exogenous drivers (climatic, anthropogenic, tectonic) of erosion across the globe. Here we leverage the expanding availability and coverage of cosmogenic-derived erosion data and historical archives of suspended sediment yield to explore these controls more broadly and place them in the context of classical geomorphic theory. We find: 1) A
relationship between mean annual precipitation/aridity and both long- and short-term erosion rates similar to that proposed in classic geomorphic literature on erosion; 2) Agricultural activities have apparently increased short-term erosion rates, outpacing natural drivers; 3) Short-term erosion rates exceed long-term rates in all climatic regions except in mid- and high-latitudes, where long-terms rates are higher due to the influence of repeated glacial cycles; 4) Tectonically active margins have generally higher long-term erosion rates and apparently lower rainfall thresholds for erosion that potentially arise due to steeper
slopes and associated landslides, overcoming vegetative root reinforcement. These results highlight the complex interplay of external controls on land surface processes and reinforce the view that timescale of observation may reveal different erosion rates and principal controls.

## 1 Introduction

Drainage basin erosion rates reflect an averaged timescale of landscape evolution in response to different possible forcing
mechanisms. However, the controls of climate and anthropogenic activities on erosion over different timescales are not well

understood. Despite impressive collections of an increasing number of long- and short-term erosion rates for drainage basins across the globe, the influences of the key external controls on basin-averaged erosion rates remains equivocal. Here we leverage existing databases of short-term sediment yield data and long-term cosmogenic radionuclides to explore the relative importance of exogenous variables including climate, anthropogenic activities, and tectonics, as well as several endogenous

drainage basin morphometrics in influencing erosion rates around the globe. This analysis has many caveats, since we employ a compilation of previously published datasets, each with its own study objectives, measurement resolutions, potential biases and uncertainties, and regional idiosyncrasies. However, we suggest that an analysis of existing global data, categorised, or filtered, using masks based on climate, tectonic, or anthropogenic activity, may yield new insights into controls on erosion and thus on landscape evolution beyond what has been shown from local or regional analyses.

**1.1 Theoretical context: Climate, tectonics, glaciers, humans**

Exploration of data generated by sediment flux monitoring programmes has revealed insights into the relationships between climatic drivers and short-term sediment yields. For example, Langbein and Schumm (1958) used a limited dataset of sediment yields to identify a relationship between sediment yield and effective mean annual precipitation (MAP) across various biomes in the USA, revealing an erosion peak in the semi-arid rainfall category. They interpreted this result by suggesting that at low

MAP there is also sparse vegetation, so erosion increases commensurately with rainfall via Hortonian overland flow (Horton, 1945). However, they posit that with sufficient rainfall, vegetation cover also increases, which at some rainfall threshold retards erosion rates because of increased root reinforcement, rainfall interception, higher infiltration, and correspondingly higher evapotranspiration and/or subsurface storm flow (Dunne and Leopold, 1978). Thus, humid regions have lower sediment yields than semi-arid landscapes, despite the higher MAP. Subsequently, Walling and Kleo (1979) extended this analysis to include

sediment data from around the globe, restricting the data to basins < 10,000 km$^2$ to minimise the effects of sediment storage, and including regions with higher MAP than the USA. Their results loosely corroborate the 1958 study, reinforcing the suggestion that sediment yields peak in dry sub-humid regions, and then apparently peak again in more humid environments. These authors suggest that intense precipitation in very humid environments may increase the weathering rate in a manner that exceeds the protection capacity of vegetation cover, leading to a rise of sediment yields. Notably, both papers that analysed

short-term sediment yield data put forth reasonable mechanistic arguments, but they are based on either limited data (Langbein and Schumm, 1958) or a 'subjectively fitted curve' through a broad scatter of grouped data (Walling and Kleo, 1979). A broadly similar relationship between MAP and erosion rate was observed in an analysis of global long-term erosion rates based on a compilation of beryllium-10 ($^{10}$Be) measurements ($n = 1,790$) characterised by an increase in erosion rate to a local maximum MAP at ~ 1,000 mm, followed by a slight reduction up to MAP of ~ 2,200 mm, and subsequently a return to

increasing values for higher MAP (Mishra et al., 2019).

There have been several prior comparisons of short- and long-term erosion rates in specific regions (Clapp et al., 2000; Kirchner et al., 2001; Gellis et al., 2004; von Blanckenburg, 2006; Kemp et al., 2020), but the findings of these studies vary

dramatically by location. Some find that short-term erosion rates are higher than long-term rates (e.g., Gellis et al., 2004) and others find the opposite result (e.g., Kirchner et al., 2001). It remains an open question how short-term and long-term erosion rates compare within the same region systematically across the globe and whether climate can be invoked to explain their differences and to identify which should be higher or lower within a given region.

Tectonics is well known to impact erosion through the links between high uplift rates, increased relief, threshold slopes, and landsliding, yet there are outstanding questions about the interplay between tectonics and climate (Ahnert, 1970; Molnar and England, 1990; Hovius et al., 2000; Whipple, 2009; Larsen et al., 2010; Larsen and Montgomery, 2012; Adams et al., 2020). A primary effect of tectonic uplift on erosion is that sediment generated in zones of high uplift (e.g., through intensive weathering) is efficiently delivered to channels in quasi-equilibrium with uplift due to the lack of accommodation space for sediment storage. Numerous studies have shown that erosion rates are positively correlated to uplift rates, as well as other morphometrics, such as total channel relief and channel slope for both short-term (Milliman and Meade, 1983; Milliman and Syvitski, 1992; Summerfield and Hulton, 1994; Aalto et al., 2006; Syvitski and Milliman, 2007; Milliman and Farnsworth, 2011; Yizhou et al., 2014) and long-term erosion rates (Granger et al., 1996; Bierman and Caffee, 2001; Schaller et al., 2001; von Blanckenburg, 2006; Binnie et al., 2007; DiBiase et al., 2010; Portenga and Bierman, 2011; Wittmann et al., 2011; Covault et al., 2013; Codilean et al., 2014; Harel et al., 2016; Schmidt et al., 2016; Grin et al., 2018; Struck et al., 2018; Tofelde et al., 2018; Hilley et al., 2019). It is currently unclear how much short- and long-term erosion rates reflect the location of samples relative to tectonic provinces and whether a climate signal may be evident in both tectonically active and tectonically passive regions.

Past glaciations have had an obvious dramatic impact on the landscape of large areas of globe, especially in the middle to high latitudes. Glacial, periglacial, and paraglacial erosion has modified topography, stripped sediment from valley bottoms and sides as a legacy of valley glaciers, whilst alpine glaciers have eroded sediment and rock from mountains (Ganti et al., 2016; Harel et al., 2016; Cook et al., 2020; Delunel et al., 2020). Even as areal coverage of glaciers is diminishing across the globe, there are lasting effects on erosion rates within formerly glaciated landscapes, especially when averaging over longer time periods. Glacial, periglacial, and paraglacial erosion has been shown to increase long-term erosion rates in specific locations within temperate and cold regions, especially within mid- and high latitudes (Portenga and Bierman, 2011; Harel et al., 2016), but is unclear whether the imprint of past glaciation is systematically preserved within erosion rates across the globe.

Finally, humans are well acknowledged agents of erosion via construction, mining, timber harvesting, and conversion of natural vegetation to agriculture (crop and pasture), the latter of which is the most prevalent in terms of global land area (Hooke, 2000; Foley et al., 2005). Global analyses of short-term erosion rates from suspended sediment records suggest that agricultural regions have higher erosion rates compared to areas with limited anthropogenic impacts (Dedkov and Mozzherin, 1996; Montgomery, 2007; Wilkinson and McElroy, 2007; Kemp et al., 2020). However, it is unclear how the signal of

anthropogenically accelerated erosion is expressed in global sediment flux records and how erosion rates derived from short-term records compare with long-term erosion rates for the same regions.


This study aims to understand the geographic expression of long- and short-term erosion rates around the globe and to explore climatic, tectonic and anthropogenic controls on erosion rates. We specifically address the following key questions: 1) What is the overall pattern of long- and short-term erosion rates categorised by climate regimes? 2) To what extent do long-term erosion rates reflect glacial (and periglacial) processes in mid- and high-latitude regions? 3) Are previously theorised

relationships between precipitation and erosion rate applicable to both short and long timescales? 4) How do anthropogenic activities affect short-term erosion rates? 5) What role does tectonics play in shredding or enhancing the influence of climate on global short- and long-term erosion rates?

## 2 Erosion proxies

To explore spatial and temporal patterns in erosion rates, we need proxies for erosion rates that capture processes at different

timescales and sufficient data from global geographic and climatic regions. Two key proxies used to represent erosion in geomorphology are: suspended sediment yields for short-term rates ($10^0$–$10^1$ y), and in-situ cosmogenic radionuclides for long-term basin-averaged erosion rates ($10^3$–$10^6$ y). Whilst each of these proxies is associated with different assumptions and different inherent uncertainties, they are commonly used in geomorphology to investigate spatial and temporal changes in erosion in response to climatic and tectonic forcing (Clapp et al., 2001; Pan et al., 2010; Wittmann et al., 2011; Yizhou et al.,

2014), to compare erosions rates between basins (Milliman and Meade, 1983; Milliman and Syvitski, 1992; Summerfield and Hulton, 1994; Dedkov and Mozzherin, 1996; Portenga and Bierman, 2011; Harel et al., 2016), and to investigate potential drivers of erosion at different timescales (Kirchner et al., 2001; Schaller et al., 2001; Covault et al., 2013; Ganti et al., 2016; Delunel et al., 2020).

Erosion rates calculated from suspended sediment yield are calculated by measuring the sediment concentration and discharge at a gauging station over years to decades, and then converting their product into mean annual sediment flux, then to sediment yield (t ha$^{-1}$ y$^{-1}$) normalised by upstream drainage area, and subsequently to erosion rate (mm y$^{-1}$), assuming a basin-averaged soil bulk density. This method provides an averaged value of erosion rate for the upstream area that neglects the storage of sediment during transportation and only accounts for sediment transported as suspended load, which makes up the majority of

sediment export from basins around the world (Leopold et al., 1964). The method neglects any sediment transported as bedload or dissolved load. The omission of bedload and dissolved load data may underestimate basin-averaged erosion rates slightly, but these data are too scarce and unevenly distributed to meta-analyse between climate zones at the global scale. A meaningful, systematic correction of short-term erosion rates is not possible due to variations in the controls on the type of sediment load between basins. For example, the percentage of bedload to the total load tends to be higher in mountain regions and drylands

(Dedkov and Mozzherin, 1996; Singer and Dunne, 2004), but the percentage of dissolved load seems to be higher in tropical regions and lower in drylands (Milliman and Farnsworth, 2011). Previous studies estimated that the bedload typically accounts for < 10% of the total load (Milliman and Meade, 1983), and the average dissolved load is even less but with significant variation (Milliman and Farnsworth, 2011). For example, in some dryland basins, dissolved load is as low as ~ 0.2% (Alexandrov et al., 2009). Despite this limitation, suspended sediment yield provides a record of recent responses within

landscapes to climatic and/or anthropogenic forcing (Walling and Webb, 1996; Walling and Fang, 2003) and is used widely as a reliable erosion proxy.

In-situ cosmogenic radionuclides such as beryllium-10 ($^{10}$Be) and aluminium-26 ($^{26}$Al), are produced by the interaction of secondary cosmic rays with minerals in rocks and soils in the uppermost few metres of the Earth's surface. The concentration

of cosmogenic radionuclides near the surface is principally a function of the production rate, radioactive decay rate, and denudation rate (or rate of total mass loss). Therefore, the concentration of cosmogenic radionuclides in river sediments can be used for estimating basin-averaged erosion rates, and the timescale of the estimation depends on the erosion rate itself (i.e. the time taken to lower the land surface) (Brown et al., 1995; Granger et al., 1996; Granger et al., 2013; Granger and Schaller, 2014; von Blanckenburg and Willenbring, 2014). This method, when applied to riverine sediments, also provides an averaged

erosion rate that is insensitive to short-term sediment storage within the upstream basin. Furthermore, this method is more practicable in basins where the land surface has been subject to continuous exposure to cosmic rays and long-term steady erosion (i.e. where abrupt and deep erosion, and long-term burial followed by erosion are minimum) (Brown et al., 1995; Granger et al., 2013; Dosseto and Schaller, 2016; Struck et al., 2018). $^{10}$Be erosion rates average erosion over a characteristic timescale determined by the nuclide concentration divided by an average nuclide production rate. This equates roughly to the

time it takes to erode ~ 60 cm of material (Granger and Schaller, 2014). Therefore, a rock lowering rate of 1 mm y$^{-1}$ equates to a 600-year timescale, 0.1 mm y$^{-1}$ to 6,000 years, and so on. Erosion rates estimated using cosmogenic nuclides represent longer timescales than suspended sediment records ($10^3$–$10^6$ y versus $10^0$–$10^1$ y) and are therefore suitable for analysing the influences of climate and tectonics, whilst being insensitive to the influences of anthropogenic activities or recent episodic erosion events with shallow erosional depth (Brown et al., 1995; von Blanckenburg, 2006; Granger et al., 2013; Granger and

Schaller, 2014; Dosseto and Schaller, 2016).

We note several uncertainties and assumptions inherent in the use of $^{10}$Be-derived erosion rates. The main assumptions are: 1) Catchments have accumulated cosmic rays in the active layer that contributes to basin erosion as measured in a channel downstream; 2) Sediment is eroded from the near surface (i.e., minimal contribution of shielded sediments from deep-seated

landslides); and 3) Erosional processes are steady and uniform in the upstream basin. These assumptions may not hold if a catchment has been fully or partially glaciated. Despite these potential limitations, we suggest that $^{10}$Be-derived erosion data obtained from published data sources are suitable for assessing broad differences in erosion rates across landscapes between climate zones, given that the original measurements were obtained to estimate erosion rates in these glaciated basins. Finally,

we note that the timescale of $^{10}$Be-derived erosion rate depends on the erosion rate itself, so they may be averaged over glacial and/or non-glacial periods, so areas mapped as formerly glaciated regions may represent erosion rates that are glacial, periglacial, and/or paraglacial. However, previous glaciations tend to enhance sediment production and lead to greater sediment fluxes during warmer periods (Ganti et al., 2016).

We are also aware of the potential confounding influences of the 'Sadler effect', in which apparent sediment accumulation (and by association, erosion) rates are slower for longer timescales due to the episodic nature of sediment transport events and preservation (Sadler, 1981) and the shredding of environmental signals during sediment transport (Jerolmack and Paola, 2010), both of which may affect comparisons between long- and short-term erosion rates. However, it has been shown that this timescale dependency is more apparent in depositional environments integrating net accumulation at a single location (Sadler and Jerolmack, 2015). Since our compiled erosion rates were estimated from suspended sediment flux (short-term) and from $^{10}$Be concentrations within fluvial sediments (long-term), rather than from stratigraphic sections in depositional zones, our results are less likely to be biased by the Sadler effect.

In addition, since cosmogenic radionuclide-derived erosion (denudation) rates include chemical weathering, but riverine sediment flux measurements do not, we recognise this may result in potential biases when comparing long- and short-term erosion in this manner. However, we expect chemical weathering to be a minor component of total denudation in most landscapes, thus minimising this potential bias between short- and long-term erosion rates used in this compilation. For example, in a semi-arid catchment in Israel, dissolved load over 15 years of measurement was 0.002% of the total sediment load (Alexandrov et al., 2009). Furthermore, in a global compilation of dissolved load and suspended load data, Walling and Webb (1983) showed that dissolved loads may be only as high as 10% of total loads, but are often far lower, e.g., fractions of a percent. These authors further demonstrate that dissolved loads are typically comprised of only ~50% chemical weathering, so these already low percentage estimates of chemical weathering would be further reduced by half. Ultimately, these lines of evidence support a direct comparison between riverine sediment flux and denudation rates derived from cosmogenic radionuclides.

Finally, we acknowledge that there are some additional embedded biases inherited from of the public databases utilised here. For example, the OCTOPUS database uses the CAIRN model (Mudd et al., 2016) to determine the integration timescale based on an assumption of penetration depth of the cosmogenic radionuclides for the region of interest (Codilean et al., 2018). Given we are presenting a global metadata analysis, we simply used the denudation rates provided in OCTOPUS without making further calculations or exploring the integration timescales for each region.

 **3 Methods**

Our analysis is based on a compilation of long- and short-term drainage basin erosion rates across the globe from existing databases and published literature (see Data availability). Data were stratified by exogenous controls such as climate, past glaciation, anthropogenic influence, and tectonics, as well as by endogenous basin morphometrics such as basin topography and basin area. We emphasise the influence of exogenous controls in this study to explore whether climatic, anthropogenic, 200 and/or tectonic influences on short- versus long-term erosion rates are detectable at the global scale. Long-term erosion rates were obtained from the Open Cosmogenic Isotope and Luminescence Database (OCTOPUS, https://earth.uow.edu.au/), which reports basin-averaged erosion rates derived from cosmogenic nuclides ($^{10}$Be and $^{26}$Al) and luminescence measurements in fluvial sediments (Codilean et al., 2018). This database classifies data based on the methods, regions, and degree of completeness. To gain the highest reliability and consistency, we only included $^{10}$Be-derived erosion rates of CRN (cosmogenic 205 radionuclide) International and CRN Australia categories from the database, resulting in a total of 3,074 data points (Fig. 1). For each data point, we extracted the erosion rate, coordinates, and drainage basin area.

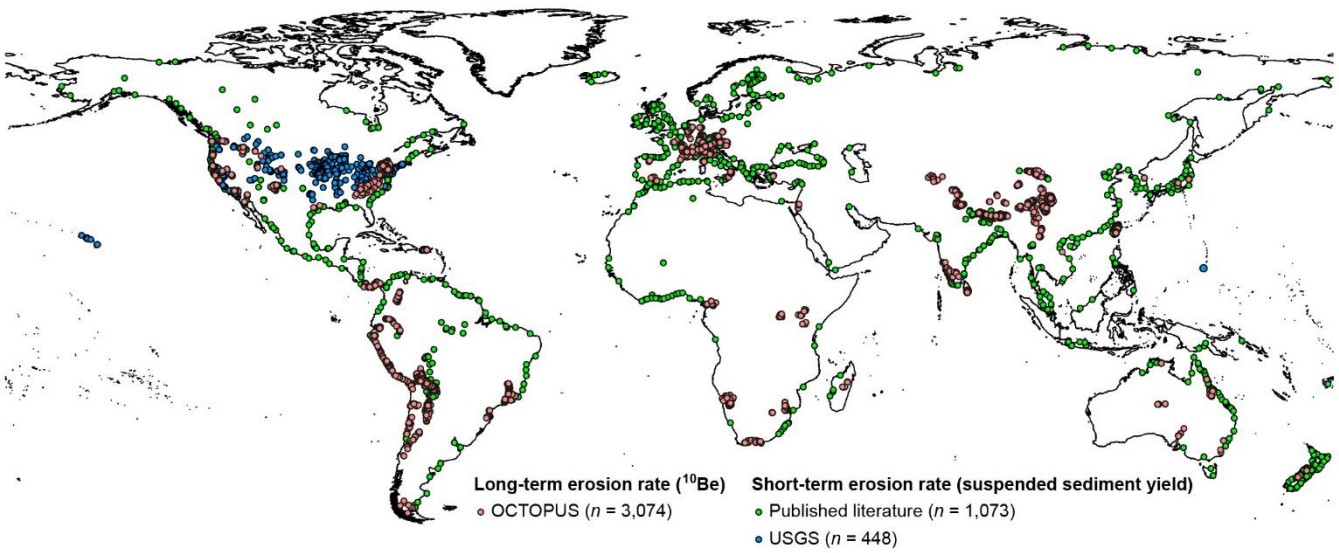

**Figure 1: Global map of drainage basin erosion rate locations. Long-term erosion rates were obtained from OCTOPUS (Open** 210 **Cosmogenic Isotope and Luminescence Database, red), estimated by $^{10}$Be in the fluvial sediments. Short-term erosion rates were compiled from published literature (green) and USGS (blue), determined by suspended sediment yield of gauging stations. Coastline is from Nature Earth (https://www.naturalearthdata.com) in the Pseudo Plate Carrée map projection.**

Short-term erosion rates were compiled from published studies and the US Geological Survey (USGS National Water 215 Information System, https://waterdata.usgs.gov/nwis), based on estimates of suspended sediment yields at gauging stations (see Data availability). From these published studies, we compiled sediment yields (t ha$^{-1}$ y$^{-1}$) at each data point. To convert sediment yields to erosion rates (mm ky$^{-1}$), we assumed a uniform sediment density of 1.6 g cm$^{-3}$ (= 1.6 t m$^{-3}$). As stated earlier,

our calculation does not consider bedload and dissolved load which typically represent a small percentage of denudation. Using this density, sediments with a depth of 0.1 mm across an area of 1 ha, have a mass of 1.6 t. A sediment yield of 1 t ha$^{-1}$ y$^{-1}$, for

example, is equivalent to an erosion rate of 0.0625 mm y$^{-1}$ (or 62.5 mm ky$^{-1}$). If data from the same gauging station were reported in multiple literature sources, we only included the erosion rate based on the most recently published data record. For USGS data, two criteria were set for choosing gauging station data: 1) Monitoring time period > 5 years, and 2) Basin area < 2,500 km$^2$. The reason for the area threshold in USGS data is to compensate for the generally larger basin sizes in the non-USGS datasets and to enable comparison to the long-term erosion rates (i.e., from the OCTOPUS database), which were

typically obtained from smaller drainage basins. Note that some of the gauging stations meeting these criteria are on the same river. We extracted the daily sediment discharge (t d$^{-1}$), converted this into sediment yield (t ha$^{-1}$ y$^{-1}$) by summing the daily data and dividing by the number of years and basin area. The sediment yield was then converted into an erosion rate for comparison with long-term erosion rates.

The USGS data are quality checked before being released by that organisation, but suspended sediment yield data compiled from peer-reviewed literature cannot be quality controlled for consistency. Therefore, uncertainty ranges will be highly variable for several reasons (Milliman and Farnsworth, 2011): the variety of measuring techniques over different periods of time; inadequate monitoring period (i.e. several rivers with historic records < 5 years); watershed modification (e.g. resulting from dam construction or climate change); variable sediment densities across basins; and potentially erroneous transcription

of the data. We have tried to reduce data uncertainties as far as possible by focusing on published sediment flux values from studies that contain descriptions of data quality control. In total, we obtained 1,521 short-term erosion rates; 1,073 from published studies and 448 from USGS (Fig. 1), with corresponding station coordinates and drainage basin areas (see Data availability).

We use two climate classifications in our analysis of global short- and long-term erosion data: 1) The Köppen–Geiger (K–G) climate classification, which is based on biome types defined by temperature and precipitation thresholds; 2) The Aridity Index (AI) is a quantitative metric for characterising the average water balance, calculated by dividing MAP by mean annual potential evapotranspiration (PET) from the Global Aridity and PET Database (Trabucco and Zomer, 2009). Here we adopt the most updated version of K–G (Peel et al., 2007), which includes five main zones (Tropical, Arid, Temperate, Cold, and Polar) and

29 sub-zones. We classified erosion rates into the main K–G zones to provide sufficient data points in each category, but we excluded the Polar zone because there are too few data. For ease of statistical comparison, we adopted the commonly used categorical approach with the following thresholds for AI classes (United Nations Environment Programme, 1997): Hyper-arid (< 0.03), Arid (0.03–0.2), Semi-arid (0.2–0.5), Dry sub-humid (0.5–0.65), and Humid (> 0.65). Note that AI provides granularity for the dryland zones (AI < 0.65) and Humid regions are lumped into one category.


For comparison with earlier studies, we explore variation in long- and short-term erosion rates against MAP derived from the GPCC Climatology Version 2020, from the Global Precipitation Climatology Centre (GPCC; https://climatedataguide.ucar.edu/climate-data/gpcc-global-precipitation-climatology-centre). This product includes monthly data covering the records from 1951 to 2000 at the globe (each gauging station has at least 10-year record) in raster format at the spatial resolution of 0.25 degree (~ 28 km at the equator). We summed the data of each grid cell to convert monthly data into yearly data and calculated the MAP for all locations where we have erosion rates. To account for any bias in the data caused by a potentially higher number of CRN samples from smaller basins than large ones, we also calculated the median erosion rate for each basin and compared these median rates against basin-averaged MAP. In using MAP to analyse relationships with long-term erosion rates, we make the assumption that whilst MAP may have changed over time, sampling locations have not shifted in their climate classification (K–G and AI) over the erosion timescales.

We investigated the influence of tectonics on erosion rates based on mapped seismicity from the Global Earthquake Model (GEM) Global Seismic Hazard Map (Pagani et al., 2018). This dataset is derived from peak ground acceleration data and highlights areas that lie within tectonically active margins, conservatively assuming > 0.08 g as the threshold in peak ground acceleration for tectonic basins. We use this map to separate tectonically active (hereafter referred to as 'tectonic') areas from non-tectonically active (subsequently referred to as 'non-tectonic') areas to support direct comparison between these categories. We recognise that this dataset is only a broad indicator of tectonics, and we use it to distinguish between areas where we might expect there to be high uplift rates (with a higher likelihood of steep, threshold slope conditions and thus high erosion) from those that are more tectonically quiescent. Essentially, we are dividing the land surface into regions where we expect high uplift and climate to be expressed in erosion signals from those where we expect climate only to dominate erosion.

To explore the influence of glacial history on erosion, we defined the spatial extent of those regions subjected to some unknown combination of glacial, periglacial, and paraglacial processes using a global vegetation map at the LGM (25,000–15,000 BP) based on fossil and sedimentary information, and expert consultation (Ray and Adams, 2001). From this dataset, we classified these regions based on: tundra; steppe-tundra; polar and alpine desert; alpine tundra; ice sheet and other permanent ice. We consider this extent to be a reasonable conservative estimate of the area influenced by glacial, periglacial, and paraglacial processes for the past few glacial cycles (i.e. to characterise the applicable range of timescales for [10]Be-derived erosion rates), although the influence of past glaciations on erosion was probably more widespread.

Anthropogenically impacted regions were determined from Foley et al. (2005), which provides global maps of croplands, and pastures and rangelands classified by the relative percentages of areas within these land uses. These maps were modified from previous studies (Ramankutty and Foley, 1999; Asner et al., 2004), in which they classified land use types from satellite images using GIS analysis. We conservatively defined anthropogenic regions with > 50% area of croplands or pastures and rangelands. We acknowledge that this is a crude classification for anthropogenic activity that may affect erosion and many other land-use

types may impact erosion rates (e.g., deforestation, mining, etc). Nevertheless, this global dataset offers the possibility to examine the specific influence of agriculture on basin erosion rates. We compared the erosion rates in classified 'croplands', and 'pastures and rangelands' (from Foley et al., 2005), against erosion rates in regions with no evidence of anthropogenic disturbance in land use.

Finally, we explored the influence of endogenous (interdependent with erosion) variables such as drainage basin area, mean channel slope, and total channel relief on erosion rates. Drainage basin areas were obtained from the OCTOPUS database, published studies, and USGS data corresponding to the erosion rate data, whilst the topographic data were extracted from the GLoPro database (Chen et al., 2019). GLoPro includes river longitudinal profiles around the globe which were extracted from NASA's 30 m Shuttle Radar Topography Mission Digital Elevation Model (SRTM–DEM). The rivers in the database are the

mainstem rivers (the longest rivers) of basins or sub-basins that do not cross K–G climate sub-zones. The database contains topographic data include the concavity, elevation, flow distance, and drainage area of each river profile. To extract river profiles from the GLoPro database for comparing topographic parameters with erosion rates, we chose a subjective distance threshold of 150 m between river profiles and erosion rate sampling points (i.e. selecting river profiles which are within 150 m to the closest erosion rate point). We then calculated the mean channel gradient and total channel relief of each river

longitudinal profile for our erosion points, which is broadly representative of the topographic influences on erosion rate. We examined the influences of basin area on erosion rates using categories: $< 500$ km$^2$, $500$–$2,500$ km$^2$, and $> 2,500$ km$^2$. The area thresholds were chosen to achieve a similar number of observations within each bin and climate category. We then calculated the ratio of short- to long-term median erosion rates ($R_{S/L}$).

To analyse the statistical difference in erosion rates between climate zones, timescales, and environmental controls, we used the Kruskal–Wallis hypothesis test. The Kruskal–Wallis is a nonparametric hypothesis test that compares the values of multiple samples to determine whether they are from the same distribution, which is useful for cases where the data may not be normally distributed. The purpose here is to identify differences between categories of data rather than not to investigate complex relationships between environmental controls. The test was conducted by the built-in function, `kruskalwallis`, in MATLAB

R2018a. Trends were fit through both long- and short-term erosion rate data versus MAP by the LOWESS smoothing method, which uses locally weighted linear polynomial regression by neighbouring data points to smooth data (Cleveland, 1979). We fitted the regression using the built-in function, `smooth`, in MATLAB, to highlight the pattern of erosion rates. We set the LOWESS polynomial as 'linear', the span as '30% of data points', and the robust option as 'off', which show the trends more clearly, although different options do not influence the results. We also provide the uncertainty range based on the coefficient

of variation of erosion rates in each 100-mm MAP bin.

## 4 Results

### 4.1 Climate influence on long- and short-term erosion rates

We first interrogate the influence of climate on the global dataset by pooling all data within each K–G or AI category and without filtering for other exogenous controls such as tectonics or land-use change. Results show that short-term erosion rates are significantly higher ($P < 0.05$) than long-term rates in all climate zones, except for the Cold K–G zone (Fig. 2, Table 1a). Within the AI categories, there is a general pattern of increasing difference between long- and short-term erosion rates with higher aridity. However, these differences are only significant for the Arid and Semi-arid categories ($P < 0.05$, Fig. 2b, Table 1b).

For the long-term erosion rates, Tropical and Arid K–G zones have significantly ($P < 0.01$) lower erosion rates (medians = 29.7 and 32.2 mm ky$^{-1}$, respectively) than Temperate and Cold zones (medians = 92.9 and 92.5 mm ky$^{-1}$, respectively, Fig. 2a, Table 1a). Within AI categories, long-term erosion rates are significantly lower in drier regions (i.e. Hyper-arid, Arid, and Semi-arid group of categories) compared to more humid regions (i.e. Dry sub-humid and Humid group of categories, $P < 0.01$) (Fig. 2b), and there are no significant differences between rates within these drier categories ($P > 0.05$, Table 1b). The maximum long-term erosion rates occur in the Temperate and Cold K–G categories and in the Dry sub-humid AI category. The number of short-term and long-term data points that are co-located is small relative to the entire dataset ($n = 79$), but the patterns are consistent (Fig. 3).

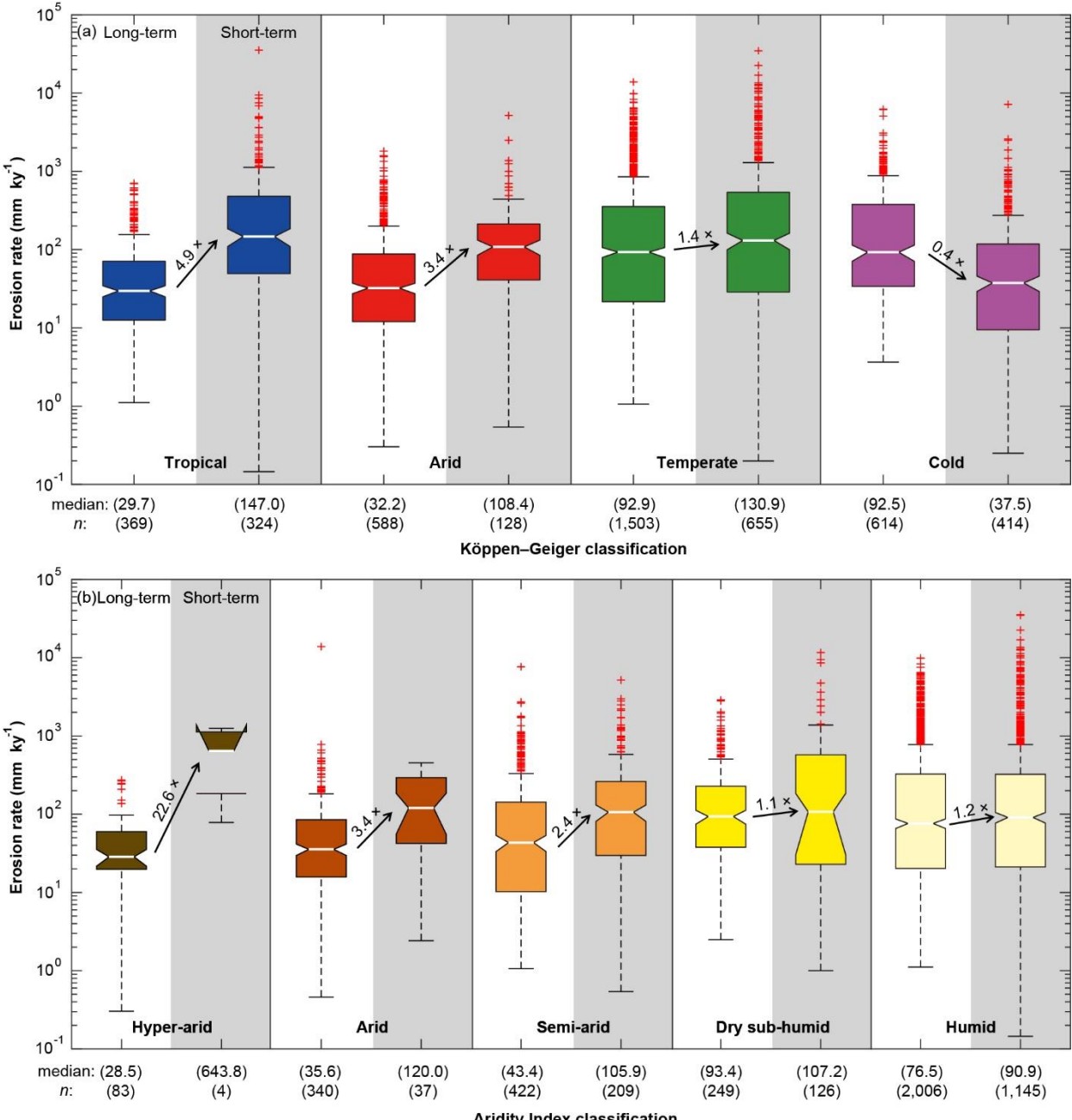

**Figure 2: Long- and short-term erosion rates (all data) for climate zones of Köppen–Geiger climate classification (a) and Aridity Index classification (b). Boxplots with white backgrounds contain the long-term rates, whilst those with the grey backgrounds contain short-term rates. For each box, the central line indicates the median value, and the bottom and top edges indicate the 25th and 75th percentiles, respectively. The notch represents the range of the median at the 95% significant level (note that the lower notch of short-term erosion rates of Hyper-arid category extends beyond the range of y-axis due to the limited number of samples in this category). Red crosses represent outliers. The arrows and numbers between boxplots in each climate zone indicate the trends and ratios of median values for short- to long-term rates ($R_{S/L}$). Median values and the number of data points for each distribution are listed below the x-axis.**

**Table 1:** *P*-values of Kruskal–Wallis tests comparing long-term ($n = 3{,}074$) and short-term ($n = 1{,}521$) erosion rates between climate zones of Köppen–Geiger climate classification (a) and Aridity Index classification (b), and between long- and short-term erosion rates of each climate zone. Bold numbers indicate significant *P*-values ($< 0.05$). The number of data points for each climate zone is listed in Fig. 2.

(a)

| Long-term rates comparison | | | | Short-term rates comparison | | | |
| --- | --- | --- | --- | --- | --- | --- | --- |
| | Arid | Temperate | Cold | | Arid | Temperate | Cold |
| Tropical | 0.88 | **<0.001** | **<0.001** | Tropical | 0.42 | 0.95 | **<0.001** |
| Arid | | **<0.001** | **<0.001** | Arid | | 0.82 | **<0.001** |
| Temperate | | | 0.54 | Temperate | | | **<0.001** |

| Long- and short-term rates comparison | |
| --- | --- |
| Tropical | **<0.001** |
| Arid | **<0.001** |
| Temperate | **0.02** |
| Cold | **<0.001** |

(b)

| Long-term rates comparison | | | | | Short-term rates comparison | | | |
| --- | --- | --- | --- | --- | --- | --- | --- | --- |
| | Arid | Semi-arid | Dry sub-humid | Humid | | Arid | Semi-arid | Dry sub-humid | Humid |
| Hyper-arid | 0.97 | 0.40 | **<0.001** | **<0.001** | Hyper-arid | 0.88 | 0.79 | 0.88 | 0.73 |
| Arid | | 0.75 | **<0.001** | **<0.001** | Arid | | 1 | 1 | 1 |
| Semi-arid | | | **<0.001** | **<0.001** | Semi-arid | | | 1 | 1 |
| Dry sub-humid | | | | 0.53 | Dry sub-humid | | | | 0.95 |

| Long- and short-term rates comparison | |
| --- | --- |
| Hyper-arid | 0.07 |
| Arid | **0.02** |
| Semi-arid | **<0.001** |
| Dry sub-humid | 1 |
| Humid | 1 |

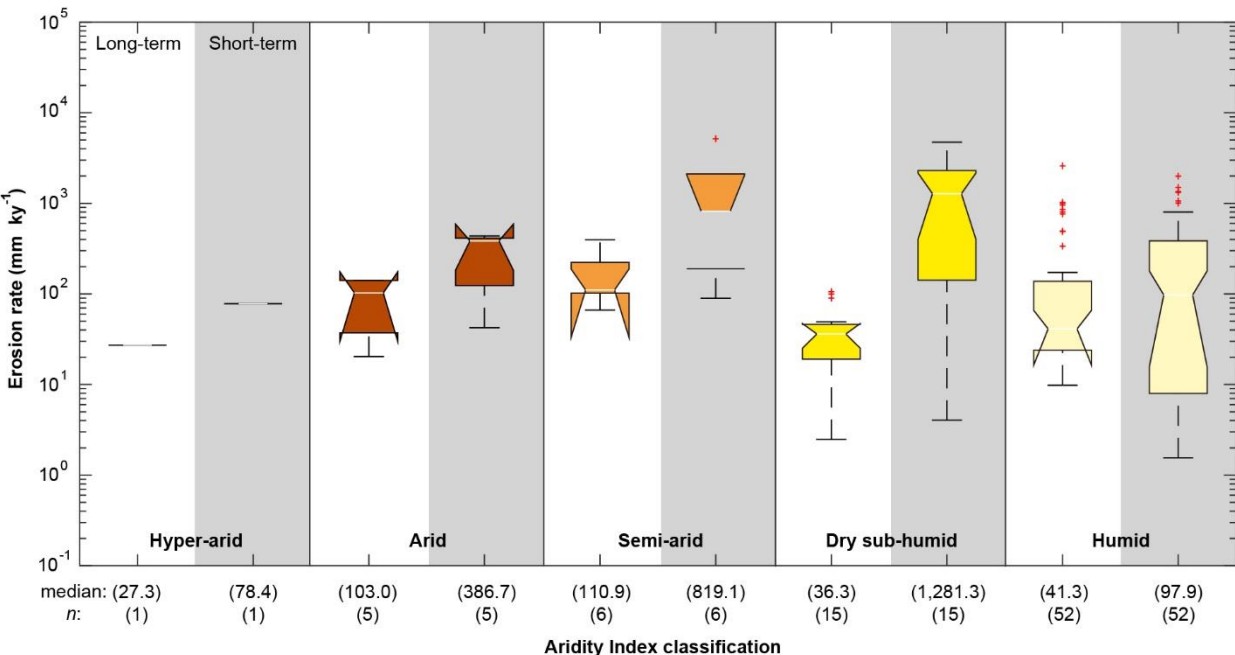

**Figure 3: Long- and short-term erosion rates at co-located points classified by Aridity Index. Note that *n* = 79 is substantially smaller than the whole dataset (Fig. 2). However, the patterns evident in Fig. 2 are upheld in this limited co-located dataset.**

The relationships between global long-term erosion rates and MAP illustrate a similar pattern to that shown for between global long-term erosion rates and AI (Fig. 2b), with the highest erosion rates expressed in the Dry sub-humid category (MAP ~ 800 mm, Fig. 4a), followed by a dip around 1,250 mm and a subsequent increase again in erosion rates in more humid regions (MAP > 1,300 mm). We verified that the effect of having multiple samples within a basin does not affect the relationships between MAP and either long- or short-term erosion rates within the global dataset (Fig. A1).

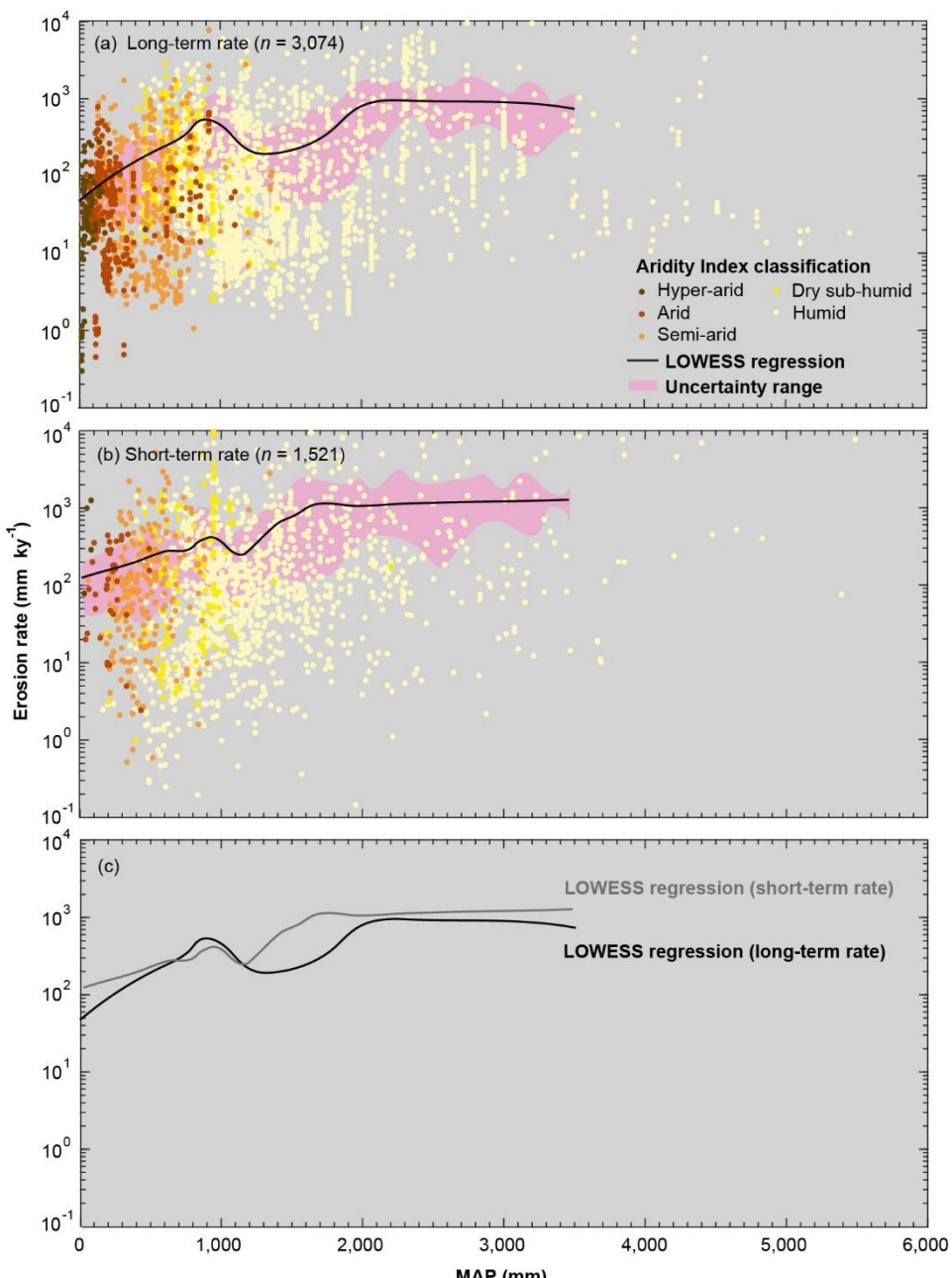

**Figure 4: Relationships between mean annual precipitation (MAP) and: (a) long-term erosion rates, and (b) short-term erosion rates. Points are colour coded by Aridity Index categories. Black curve in each panel is LOWESS regression, and the pink shading represents the uncertainty of the regression. (c) The comparison of LOWESS regressions between long- and short-term erosion rates. The regressions are truncated at extremely humid regions (MAP > 3,500 mm) owing to limited number of data points in these regions.**

Within the short-term erosion rates, distributions based on box plots indicate that there is no apparent dependency on climate according to either climate classifications ($P > 0.05$), except in the Cold zone of K–G classification, where there were significantly lower erosion rates compared to other climate zones ($P < 0.01$, Fig. 2, Table 1). The medians of short-term erosion rates in all climates are generally between 90 and 150 mm ky$^{-1}$, whereas the Cold K–G zone median is only 37.5 mm ky$^{-1}$, and the Hyper-arid AI category is as high as 643.8 mm ky$^{-1}$ (note that the result of Hyper-arid category may not be robust because of limited available data). However, the relationship between short-term erosion rates and MAP shows an overall positive relationship (Fig. 4b) and displays a similar pattern as Fig. 4a, of peaking (at 920 mm), dipping and then peaking again (at 1,750 mm), albeit the overall short-term rates are consistently higher than for the long-term. Fig. 4c highlights a remarkable similarity in the shape of the short- and long-term erosion rate LOWESS curves, despite the two erosion datasets being based on completely different and independent erosion proxies.

## 4.2 Influence of tectonics

With respect to tectonics, we found generally higher erosion rates (both short- and long-term) for tectonic sites compared to non-tectonic sites (Figs. 5, A2, A3). We also note that the average (purple) LOWESS curve for the entire dataset plots close to that for the tectonic curve (note the log scale), especially for the long-term rates in which there is an approximate 2:1 ratio of tectonic to non-tectonic points (Fig. 5a). When we look at differences in erosion rates based on our tectonic classification across MAP, and AI and K–G climate classes, we observe: 1) Overall higher long-term erosion rates in tectonic basins (Fig. A2); 2) Tectonic basins have mostly higher short-term rates except in the most arid regions, where the differences between tectonic and non-tectonic locations are not significant (Fig. A3); 3) A general increase in erosion rates with MAP for both tectonic and non-tectonic locations that mostly levels out at MAP > 2,000 mm (Fig. 5); and 4) A consistent increase in long-term erosion rates with humidity (AI) for both tectonic and non-tectonic sites (Fig. A2b).

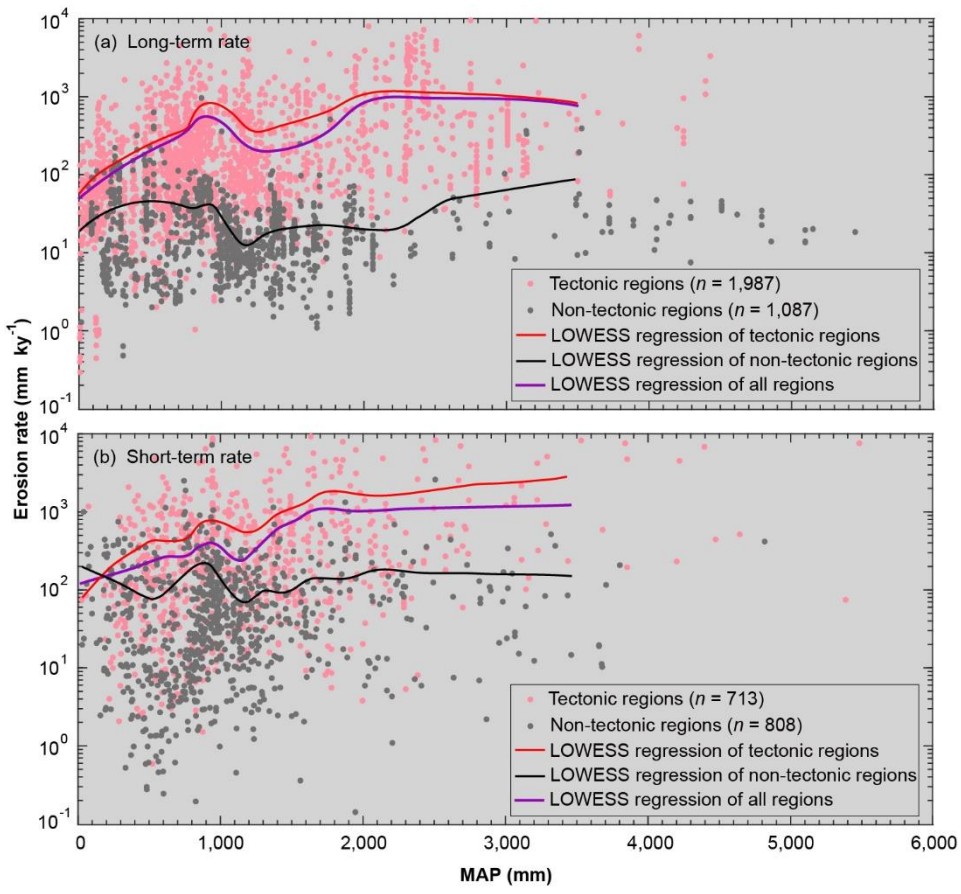

**Figure 5: Relationships between mean annual precipitation (MAP) and long- (a) and short-term (b) erosion rates split into tectonic (pink) and non-tectonic (grey) regions. Red curve is LOWESS regression for erosion versus MAP in tectonic regions, black curve is LOWESS regression for erosion versus MAP in non-tectonic regions, and purple curve is LOWESS regression for erosion versus MAP across all regions. The regressions are truncated at extremely humid regions (MAP > 3,500 mm) owing to limited number of data in these regions.**

## 4.3 Influence of glaciation on long-term erosion rates

To explore the influence of past glaciations on long-term erosion rates, we compared data for those locations that are currently in the Temperate K–G zone and were previously in glacial and periglacial zones during the Pleistocene (e.g. north-western Europe, part of the Andes, the Himalayas, and New Zealand) against the Temperate sites that were not glaciated at the LGM (Fig. 6), based on the work of Ray and Adams (2001). We find that the median long-term erosion rate for formerly glaciated regions of the Temperature zone is approximately 5 times higher than in non-glaciated regions (medians = 202.3 and 41.4 mm ky$^{-1}$, respectively, $P < 0.01$). This result supports the role of glacial and periglacial influences, such as basal erosion, freeze–thaw, weathering processes, etc, in shaping surface across the landscape resulting in higher long-term erosion rates.

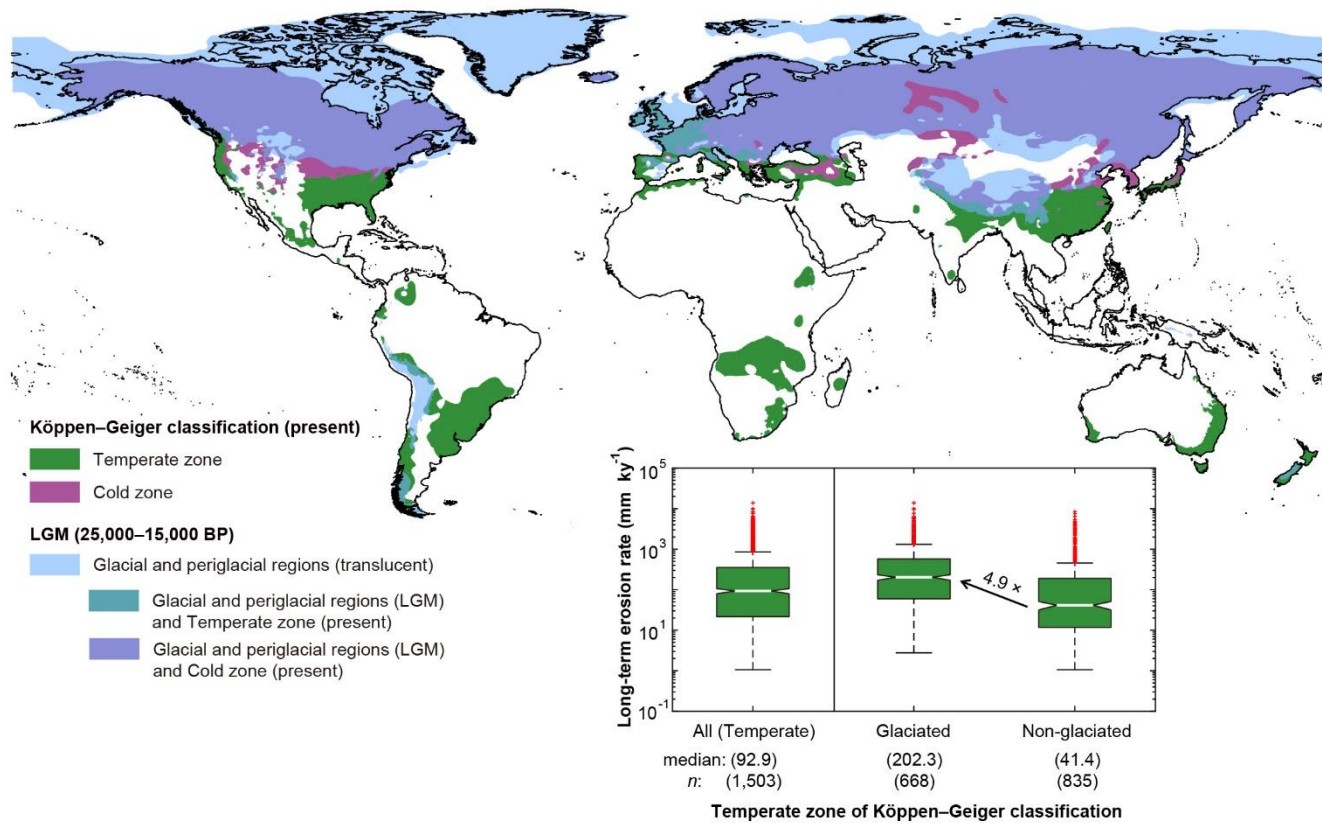

**Figure 6: The extent of glacial and periglacial regions at the last glacial maximum (LGM) and the area of Temperate and Cold zones of Köppen–Geiger climate classification in the present. The glacial and periglacial regions were drawn from Ray and Adams (2001), according to the description in Methods. The inset panel compares long-term erosion rates in the Temperate K–G zone with and without glacial influences at the LGM, indicating 4.9 times higher median erosion rates in formerly glaciated regions compared to**
410 **non-glaciated regions.**

## 4.4 Anthropogenic influences on short-term erosion rates

Anthropogenic influences on short-term erosion rates were examined using agricultural land use as a proxy, since it represents the largest anthropogenic impact in terms of global land area. The median short-term erosion rate for these agriculturally
influenced areas is 1.4 times higher than in regions without these anthropogenic influences (78.3 mm ky$^{-1}$, $P < 0.05$, Fig. 7). However, there was no significant difference in erosion rates between these two types of anthropogenically impacted land uses (104.2 and 114.0 mm ky$^{-1}$, respectively, $P > 0.05$).

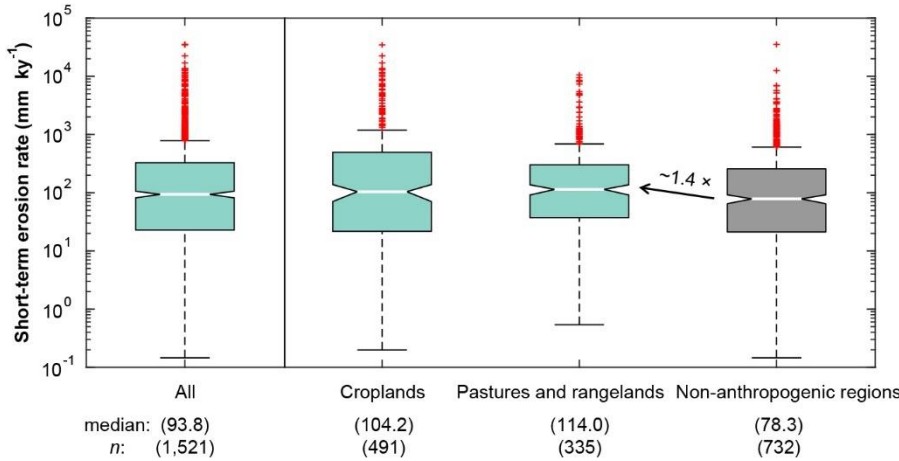

**Figure 7: The comparison of global short-term erosion rates with and without anthropogenic influences. The extent of 'croplands' and 'pastures and rangelands' were digitised from Foley et al. (2005), and the figure shows that short-term erosion rates with anthropogenic influences are ~ 1.4 times higher than in non-anthropogenically impacted regions.**

## 4.5 Influence of basin morphometrics

Given the large number of studies that link basin area and topography to erosion rates, we also investigated these endogenous controls, although we recognise that these metrics tend to be affected by both tectonics and climate due to deformation of basins by topographic uplift and by differences in runoff generating mechanisms, respectively (Ahnert, 1970; Molnar and England, 1990; Hovius et al., 2000; Whipple, 2009; Larsen et al., 2010; Larsen and Montgomery, 2012; Adams et al., 2020). Across the whole dataset, for both long- and short-term erosion rates, there is no clear relationship with basin area ($R^2$ values of power-law regressions between basin area and long- and short-term erosion rates are both < 0.01; Fig. 8). We found a negative relationship between the ratio of short- to long-term median erosion rates ($R_{S/L}$) and basin area for each K–G climate zone, except the Cold zone (Fig. 9). Generally, short-term erosion rates are several times higher than long-term rates in small basins, whilst in large basins, long-term rates tend to be more similar or even higher than short-term rates. In addition, long-term erosion rates are positively corelated to channel gradient and channel relief ($R^2$ = 0.29 and 0.24, respectively; $P < 0.01$), whilst for short-term erosion rates, the influences of these topographic parameters are unclear (Fig. 10).

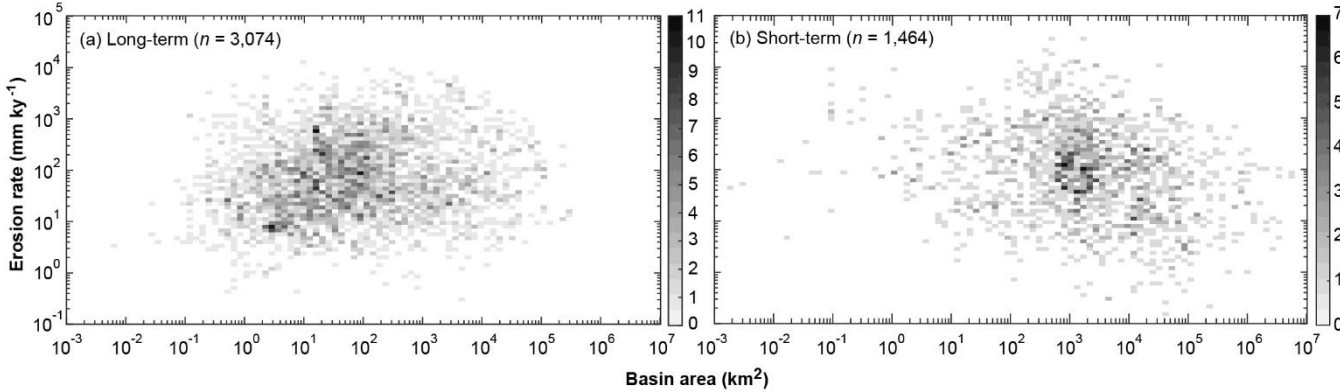

**Figure 8: Density scatter plots of the drainage basin area versus long- (a) and short-term (b) erosion rates. The colour ramp indicates the number of data points in each pixel.**

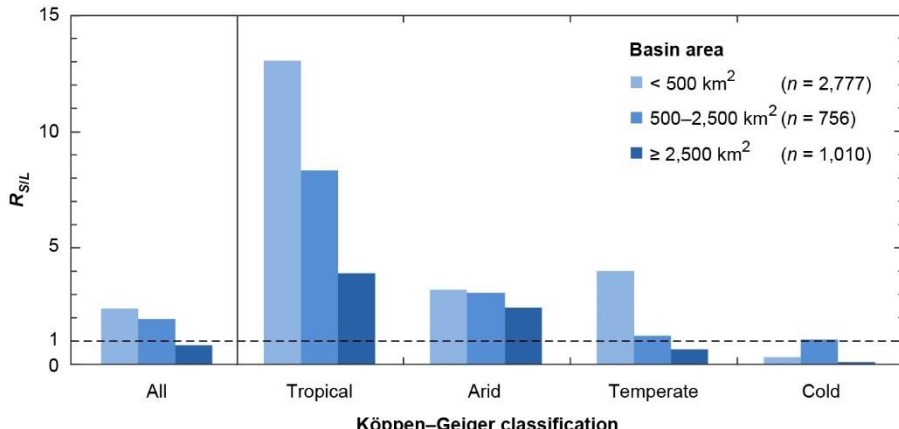


**Figure 9: The ratio of short- to long-term erosion rates ($R_{S/L}$) of each basin area bin between climate zones for the Köppen–Geiger climate classification. Each ratio was calculated from the medians of short- to long-term erosion rates of each area bin in each climate zone. The numbers of data points in each basin area bin (short-term plus long-term erosion rates) are listed in the legend. The dotted line indicates equality of short- and long-term rates. Generally, in smaller basins, short-term erosion rates tend to be higher than**
**long-term rates compared to larger basins.**

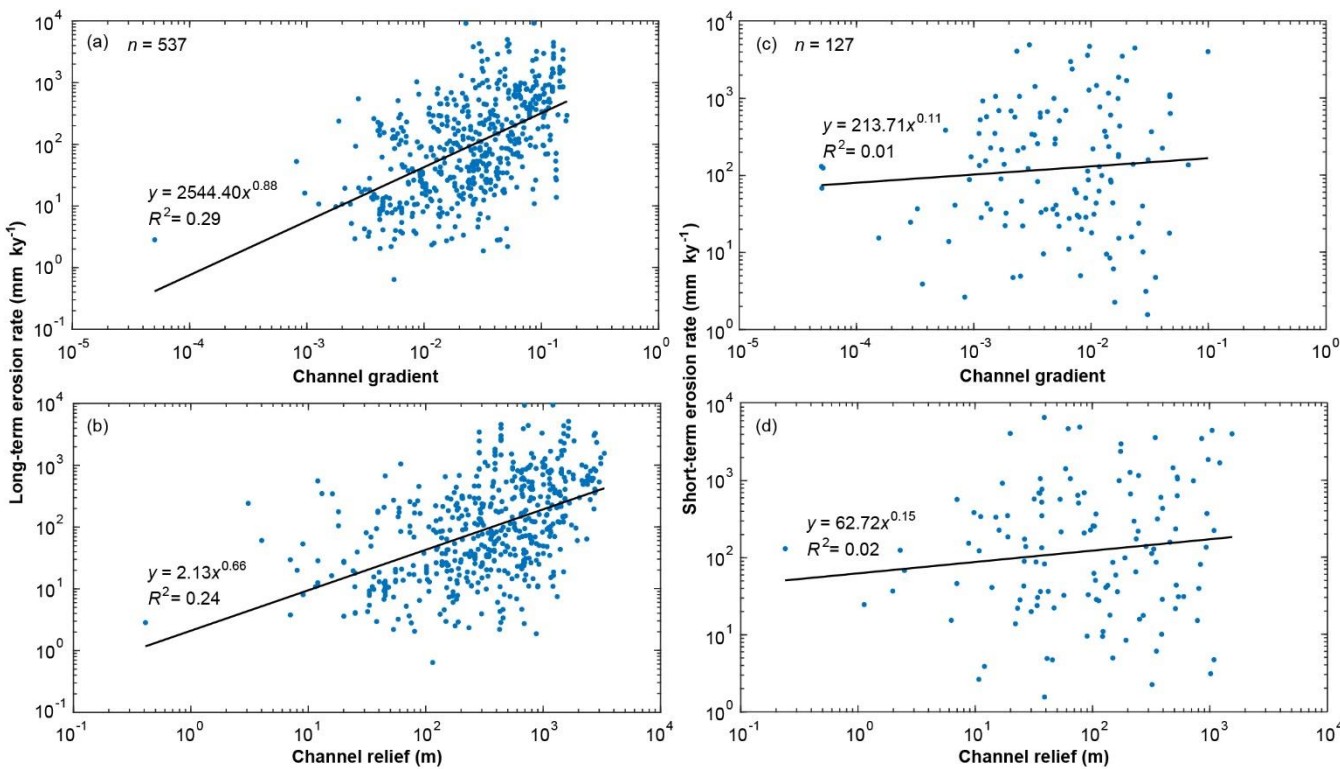

**Figure 10: Relationships between topographic parameters of river longitudinal profiles and long- (a, b) and short-term (c, d) erosion rates.**


## 5 Discussion

We set out to investigate the key potential drivers of erosion and their influence on erosion rates over short ($< 10^1$ y) and long ($10^3$–$10^6$ y) timescales, and we compared rates between these timescales for two climate classifications, as well as for various masks representing other plausible exogenous controls. We specifically investigated erosion rate variations through the lenses
of climate (classifying by Köppen–Geiger and Aridity Index classifications, mean annual precipitation, and historical maps of glaciated versus non-glaciated regions), tectonics (classified by peak ground acceleration map), anthropogenic activities (classified agricultural regions), and basin topography (channel gradient and channel relief). We fully acknowledge that drainage basin erosion rates are controlled by various (sometimes interrelated) factors, some of which may compound erosion at a particular site (e.g. high rainfall regime with intensive land use), and some of which may offset each other (e.g. agricultural
activities may accelerate erosion in lowland areas where erosion rates would otherwise be expected to be low under undisturbed conditions).

A key finding from our meta-analysis of global data is that there is a relationship between long-term erosion rates and climate (Figs. 2b, 4a), which broadly corroborates early theoretical work on short-term erosion rates from sediment yields (Fig. 11a;

Langbein and Schumm, 1958; Walling and Kleo, 1979) and modelling (Istanbulluoglu and Bras, 2006; Collins and Bras, 2008). Even when we stratify the global data into tectonically active and non-tectonic regions (Figs. 5; 11b), we find that tectonic regions tend to have a higher erosion peak. This suggests that landscapes in active margins are more primed for erosion, perhaps based on steeper (threshold) slopes (Ahnert, 1970; Larsen and Montgomery, 2012; Adams et al., 2020), and related factors. Furthermore, the lack of a dip in long-term erosion rates at slightly higher values of MAP, as is seen for non-tectonic

areas (Figs. 5; 11b), suggests that the inferred role of vegetation stabilisation (Schmidt et al., 2001) that has been invoked in classic literature (Langbein and Schumm, 1958) may not hold in areas where slopes are already primed for higher erosion due to coupling with landslide susceptibility (Hovius et al., 2000; Larsen et al., 2010). These data suggest that once there is sufficient input rainfall in tectonic areas, landsliding and other hillslope processes may override the influence of vegetation in stabilising the landscape and thereby reducing erosion (Fig 11b).


Regardless of tectonic activity, long-term erosion rates peak at MAP values of 870 mm (within the Dry sub-humid AI category) followed by a dip before peaking again at high MAPs (2,200 mm). This finding is consistent with early theoretical work that proposed that sediment yields peak in semi-arid regions due to the combination of rainfall (high enough) and vegetation cover (low enough) that results in optimum conditions for erosion (Fig. 11a) (Langbein and Schumm, 1958). Note that for direct

comparison with other data, we have replotted the original Langbein-Schumm curve converting their effective precipitation values (determined based on runoff) to MAP by assuming 50% losses (0.5 runoff coefficient) of incoming precipitation, which shifts their erosion peak to the dry sub-humid precipitation regime (MAP = 500–800 mm). Following Langbein-Schumm (1958), Walling and Kleo (1979) found a similar erosion peak in dry sub-humid regions (MAP ~ 600 mm). They also identified two further peaks in sediment yield in humid regions, where precipitation may be particularly intense and weathering

(erodibility) may be high (Fig. 11a), although the authors acknowledged that their fit to data points was subjective. Based on a more limited compilation of global [10]Be data, Mishra et al. (2019) found a similar relationship between long-term erosion rate and precipitation, albeit with differences in erosion peak locations perhaps due to artefacts from their polynomial fit (Fig. 11a). Nevertheless, there is clear corroboration in data and theoretical underpinning supporting a peak in erosion rates within dry sub-humid landscapes near the transition from dry to wet precipitation regimes and sparse to extensive vegetation cover

(Figs. 2b, 4a, 11a; Langbein and Schumm, 1958; Molnar et al, 2006; Collins and Bras, 2008).

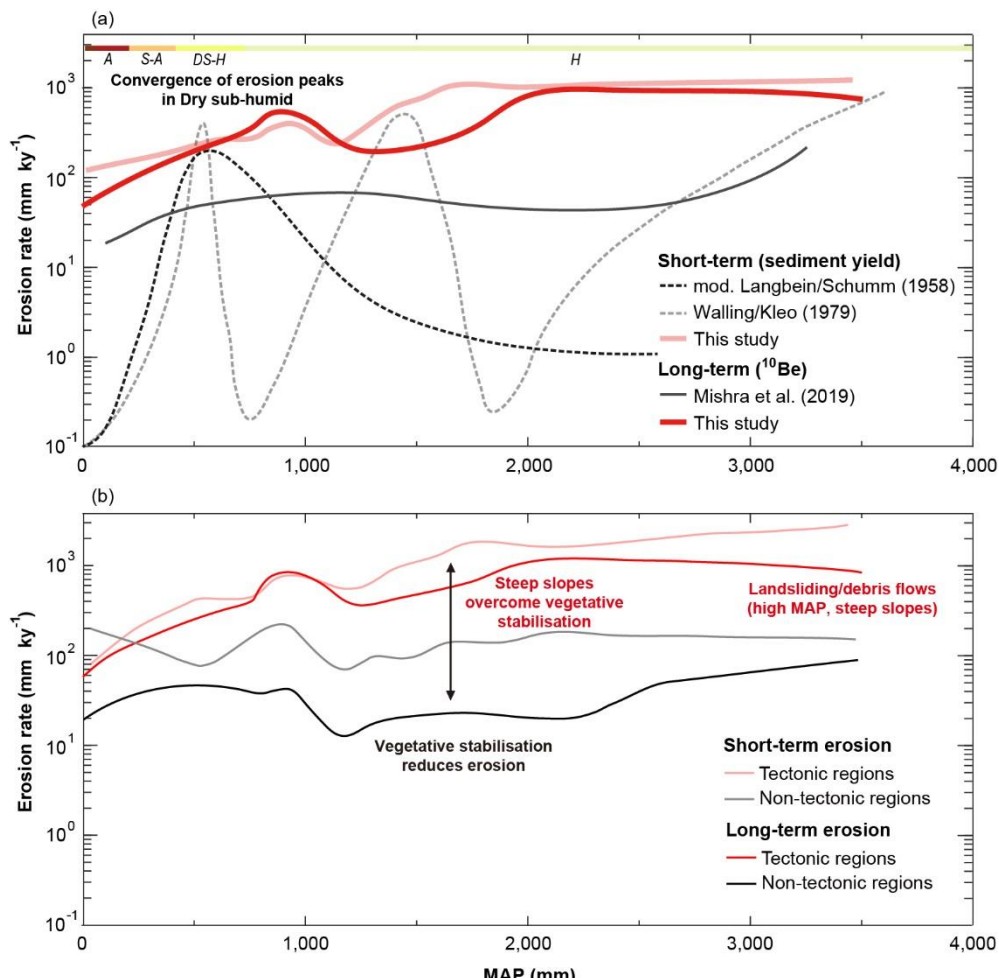

**Figure 11: (a) Synthesis of non-linear relationships between MAP and short-term erosion rates (modified Langbein and Schumm, 1958 (see text), Walling and Kleo, 1979, and this study), and between MAP and long-term erosion rates (Mishra et al., 2019, and this study). (b) Relationships short- and long-term erosion rates and MAP classified between tectonic and non-tectonic regions. MAP precipitation regimes akin to Aridity Index classes are shown along top. The figure highlights the convergence of erosion peaks in Semi-arid and Dry sub-humid regions.**

Remarkably, despite using spatially independent datasets of two distinct erosion proxies, we find a broadly similar relationship between short-term erosion rates versus MAP, albeit short-term rates are consistently higher than the long-term rates (Fig. 4b, c). Despite differences in overall erosion magnitude and a large amount of scatter in both datasets, the relationships of long- and short-term rates with MAP demonstrate an initial erosion peak in the Dry sub-humid regions, followed by a dip and then another rise towards very humid zones.

We found that short-term erosion rates are generally higher than long-term rates in all climate categories for both classifications, except for the K–G Cold zone (Fig. 2), which is mostly covered by contiguous boreal forest. The higher short-term erosion rates should be viewed through the lens of a recently more erosive environment due to the impact of humans globally. After classifying erosion rates based on land use, we found those in both croplands and pastures/rangelands to be similar and significantly higher than erosion rates for locations without anthropogenic influences (Fig. 7). These results support previous

findings that human activities significantly increase short-term erosion rates, and that they are consistently detectable around the globe. Human activities have increased short-term erosion rates by an estimated one to two orders of magnitude (Milliman and Syvitski, 1992; Dedkov and Mozzherin, 1996; Montgomery, 2007; Wilkinson and McElroy, 2007; Kemp et al., 2020), suggesting that human influences on sediment yields outweigh natural processes (Hooke, 2000; Wilkinson and McElroy, 2007; Kemp et al., 2020). Among the many anthropogenic activities expressed on surface erosion around the globe, agriculture has

one of the highest impacts on the land surface because it directly alters both vegetation through replacement of forest canopies with low-interception coverage crops, and soils through replacement of natural profiles containing developed organic layers with homogenised profiles that undergo cycles of tillage and surface compaction (Hooke, 2000). This anthropogenic disruption of vegetation and soils should create higher susceptibility to erosion by rainsplash, runoff, and wind (Dedkov and Mozzherin, 1996; Wilkinson and McElroy, 2007; Kemp et al., 2020), even in lowland environments. The eroded material would then

contribute to stream channels, where it would be measured as systematically elevated sediment yields compared to pre-historic levels.

It is worth noting that the difference in short-term erosion rates between anthropogenic and non-anthropogenic regions shown here is smaller than was shown in previous studies (Dedkov and Mozzherin, 1996; Montgomery, 2007; Wilkinson and McElroy,

2007; Kemp et al., 2020). For example, Dedkov and Mozzherin (1996) estimated that anthropogenic activities increase sediment yields by a factor of 3.5 in large rivers and a factor of 8 in small rivers. We speculate that one of the main reasons for this discrepancy is that here we may be underestimating the amount of area that is influenced by anthropogenic activity, based on our defined threshold of > 50% agricultural area. Another possibility is that our analysis may be including more short-term erosion rates sampled in anthropogenically impacted regions, where substantial soil and water conservation efforts

in upstream basins, as well as engineering structures (e.g. dams) that trap sediment may result in artificially lower sediment yields (Walling and Webb, 1996; Hooke, 2000; Walling and Fang, 2003; Syvitski et al., 2005; Singer and Dunne, 2006; Wilkinson and McElroy, 2007; Singer and Aalto, 2009).

Previous studies comparing short-term and long-term erosion rates have shown contrasting results and interpretations

depending on location (Clapp et al., 2000; Kirchner et al., 2001; Gellis et al., 2004; von Blanckenburg, 2006; Kemp et al., 2020). Our global analysis sheds some light on this debate, since we use a range of exogenous variables in the classification of erosion rates to demonstrate where you might expect short-term rates to be higher than long-term rates and vice versa. Specifically, we show that arid regions (from the K–G classification) have significantly higher short-term erosion than long-

term (Fig. 2a), and that the strength of this signal is greater with higher aridity (Fig. 2b) – a pattern which is seen in both tectonic and non-tectonic regions (Fig. A2, A3), ruling out the potential sampling bias of data. This result is consistent with prior research findings from dryland regions (Clapp et al., 2000; Gellis et al., 2004), which attribute higher short-term erosion rates to anthropogenic activities, specifically the expansion of grazing. In contrast, we show that cold regions have higher long-term rates (Fig. 2a), which are likely to have been affected by past glaciations (including paraglacial and periglacial processes) throughout the temperate zone (Fig. 6). This result is consistent with the findings of Kirchner et al., 2001, who explained their higher long-term erosion rates based on the occurrence of low-frequency, high-magnitude erosion-inducing wildfires that are generally not captured in short-term sediment yield records, similar to glaciers. Although the basins from that study are not directly within the glacial mask from Fig. 6, which predominantly represents continental glacial extent, these basins were likely subjected to Pleistocene cold spells leading to the accumulation of alpine glaciers, which may have accelerated erosion during and after melting, thus leading to higher long-term erosion rates even without the influence of wildfires.

When erosion is averaged over timescales long enough to capture the effects of past glaciations, the signal of glacial erosion appears to be detectable for mid- and high-latitude regions, wherein formerly glaciated locations within the Temperate K–G climate zone exhibit erosion rates five times higher than unglaciated regions within this same climate zone (Fig. 6). This result is consistent with previous studies, which argued that in mid- and high-latitude regions, long-term erosion rates tend to be higher than low-latitude regions because glacial, periglacial, and paraglacial processes stripped away the underlying land surface and increased physical weathering through freeze–thaw processes (Schaller et al., 2002; Portenga and Bierman, 2011; Harel et al., 2016; Cook et al., 2020). Our result of higher erosion in regions with past glaciation is also consistent with the relatively low ratio of short- to long-term erosion for the Humid AI category (Fig. 2b), which likely arises in part because the Humid class includes 46% of the total number of formerly glaciated sites included in our analysis. The strength of this glacial signal in the data suggests that the effects on long-term erosion rates are real, even if there are potential uncertainties and biases in the cosmogenic radionuclide record spanning glacial periods (Ganti et al., 2016).

Our analysis of the influence of inherent drainage basin characteristics on erosion rates addresses their roles as endogenous basin drivers of erosion (in contrast to the exogenous drivers: climate, tectonics, and anthropogenic activities). We found positive relationships between both channel gradient and total channel relief and long-term erosion rates (Fig. 10a, b), yet there was no clear relationship between short-term erosion rates and these topographic indices (Fig. 10c, d). Drainage basin steepness is considered to be a critical control on erosion rates (e.g. Summerfield and Hulton, 1994; Granger et al., 1996; Portenga and Bierman, 2011). Drainage basins with higher steepness tend to produce higher velocity of runoff because of the downslope vector of potential energy, which increases the shear stress of water flow and thus produces higher erosion that shapes land surface and transports sediments downstream (Knighton, 1998; Whipple and Tucker, 1999). In addition, steep drainage basins are often located in tectonically active regions, with low bedrock strength, high frequency of landslides (Binnie et al., 2007; Grin et al., 2018), and high precipitation rates induced by orography (Willett, 1999; Roe et al., 2002), all of which would tend

to increase erosion rates (see below). Therefore, it is logical that there would be a strong relationship between topography and erosion especially for tectonically active margins (as shown previously in many studies), yet it is less obvious why short-term rates do not exhibit this relationship. One possibility is that agriculture, a key anthropogenic influence on erosion, tends to cluster in downstream parts of drainage basins with gentler slopes (Wilkinson and McElroy, 2007). In upstream sections of drainage basins, anthropogenic activities that accelerate erosion (e.g. deforestation) may be ameliorated (from a sediment yield perspective) by soil and water conservation efforts (Montgomery, 2007), and/or by the trapping of sediment within reservoirs (Walling and Webb, 1996; Walling and Fang, 2003; Syvitski et al., 2005). Thus, sediment yields may vary substantially from upstream to downstream even within the same basin, depending on the locations of these anthropogenic activities within the landscape, as well as cycles of erosion, deposition, and remobilisation, which would lead to a scrambling of the relationship between topography and erosion (Fig. 10c, d).

We further investigated short- and long-term erosion rates categorised by basin area but found no strong relationship between basin area and long-term or short-term erosion rates within our compiled global dataset (Fig. 8). There are several factors that potentially obscure any systematic relationship between basin area and erosion including the sampling location within the basin, tectonic setting, and underlying lithology. Apparently, the effect of basin area alone on either short- or long-term erosion rates is not detectable because it is obscured by the various other controls. However, when we classified the ratio of short- to long-term erosion rates, $R_{S/L}$, by basin area, we found that this ratio is lower for larger basins, except in the Cold K–G climate zone (Fig. 9).

Prior work has shown that the differences between long- and short-term erosion rates are less discernible in large basins compared to small basins, due to the sediment buffering capacity of the former (Wittmann et al., 2011; Covault et al., 2013). Buffering capacity is determined by the balance between sediment supply and the accommodation space for deposition (Wittmann et al., 2011; Covault et al., 2013), favouring larger basins. Notably, the $R_{S/L}$ values are less sensitive to basin area within arid catchments compared to more humid zones (Fig. 9) because arid regions have a distinctive hydrological regime, where storms tend to have shorter duration, smaller spatial coverage, and high spatial variability, which generate partial area runoff (Yair et al., 1978; Singer and Michaelides, 2017; Michaelides et al., 2018). Arid regions also experience transmission losses within porous river channels, resulting in a breakdown in the relationship between basin area and streamflow, compared to the positive relationship found in humid regions (Knighton and Nanson, 1997; Tooth, 2000; Singer and Michaelides, 2014; Jaeger et al., 2017). These characteristic features of arid zone hydrology reduce the influence of basin area on hydrological processes, including sediment yields, leading to weaker buffering capacity of drainage basins in arid regions. An additional factor that may explain the lack of area control in arid regions is that short-term erosion rates tend to be systematically higher than long-term rates (Gellis et al., 2004; Bierman et al., 2005), which creates values of $R_{S/L}$ closer to unity, regardless of basin size. In tropical regions, the $R_{S/L}$ values are generally higher than other climate zones, which may result from lower long-term erosion rates compared to Temperate and Cold zones (perhaps due to the lack of past glaciation), and higher short-term erosion

rates due to intensive agricultural activity which may destroy the dense vegetation cover (e.g. deforestation), although the ratio declines substantially with basin size (Fig. 9). In the Temperature and Cold K–G zones, the $R_{S/L}$ values are generally lower for all basin area classes than the other two categories (i.e. long-term erosion rates are more similar to short-term rates, or even higher) likely because glacial and periglacial processes since the LGM led to increased long-term rates.

## 6 Conclusions

By compiling and analysing erosion rates from globally distributed sites, we demonstrate a few key differences in long- and short-term rates and their dominant controls: 1) Short-term erosion rates are significantly higher than long-term erosion rates in all climate zones except in the K–G Cold zone; 2) Long-term erosion rates are higher in mid- and high-latitude regions (including the K–G Cold zone and part of the Temperate zone), likely due to glacial, periglacial, and paraglacial processes; 3) Long-term erosion rates are systematically higher in tectonically active regions but display a similar pattern as non-tectonic regions with an erosion peak in the Dry sub-humid climate zone; 4) Both long- and short-term erosion rates are strongly related to indices of climate, tectonics and topography despite high variability in the data; 5) Short-term erosion rates are higher than long-term erosion rates likely due to human activities; and 6) Short-term erosion rates are generally several times higher than long-term rates in small basins, showing that human-induced erosion is more detectable in small basins with lower sediment buffering capacity, whilst long-term erosion rates tend to be similar or even higher than short-term rates in large basins. A key finding is that a relationship exists between long- and short-term erosion rates and climate with an erosion peak in the Semi-arid – Dry sub-humid rainfall regime, which likely reflects the balance between precipitation and vegetation cover broadly corroborating prior studies (Langbein and Schumm, 1958; Walling and Kleo, 1979). This paper does not claim to provide the definitive answers to the global controls on erosion but aims to contribute a new analysis of short- and long-term erosion rates within the context of classic geomorphic theory that we hope may provide useful perspective in the ongoing debate.

## Data availability

Short-term erosion rate data from compiled sediment fluxes are available at the University of Bristol data repository, data.bris, at https://doi.org/10.5523/bris.1pq50eh0902da25aps5nhc1ngv.

# Appendix

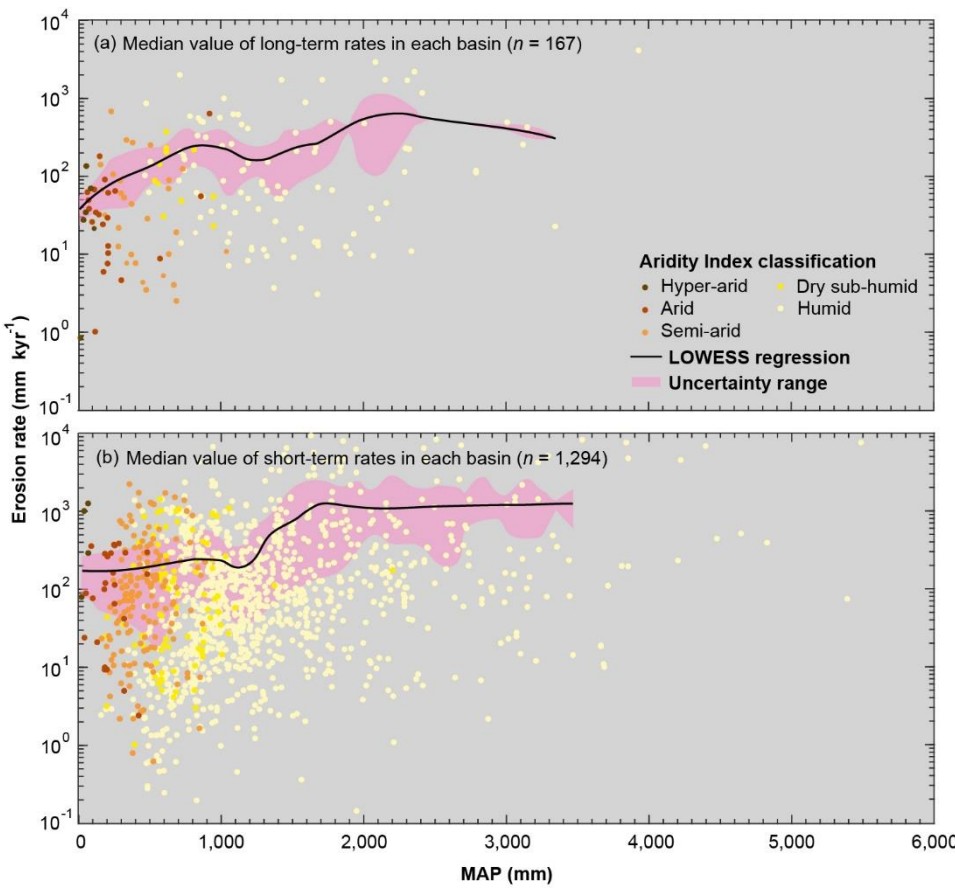

**Figure A1: Relationships between mean annual precipitation (MAP) and: (a) median value of long-term erosion rate in each basin, and (b) median value of short-term erosion rate in each basin. Points are colour coded by Aridity Index categories. Black curve in each panel is LOWESS regression, and the pink shading represents the uncertainty of the regression. The regressions are truncated at extremely humid regions (MAP > 3,500 mm) owing to limited number of data points in these regions.**

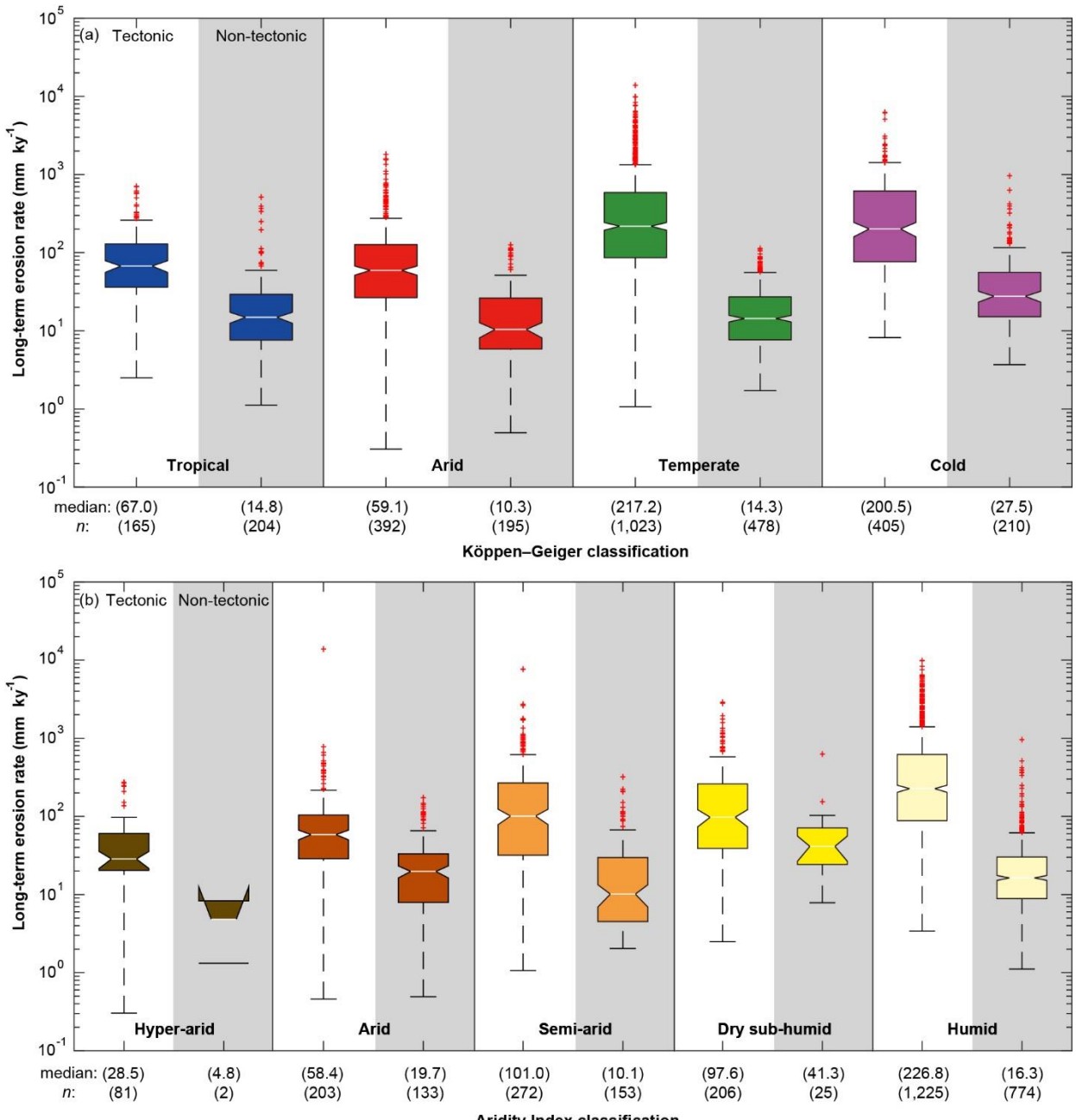

**Figure A2:** Long-term erosion rates classified by tectonic (white backgrounds) and non-tectonic regions (grey backgrounds) for climate zones of Köppen–Geiger climate classification (a) and Aridity Index classification (b). Long-term erosion rates in tectonic regions are all higher than in non-tectonic regions across all climate zones.

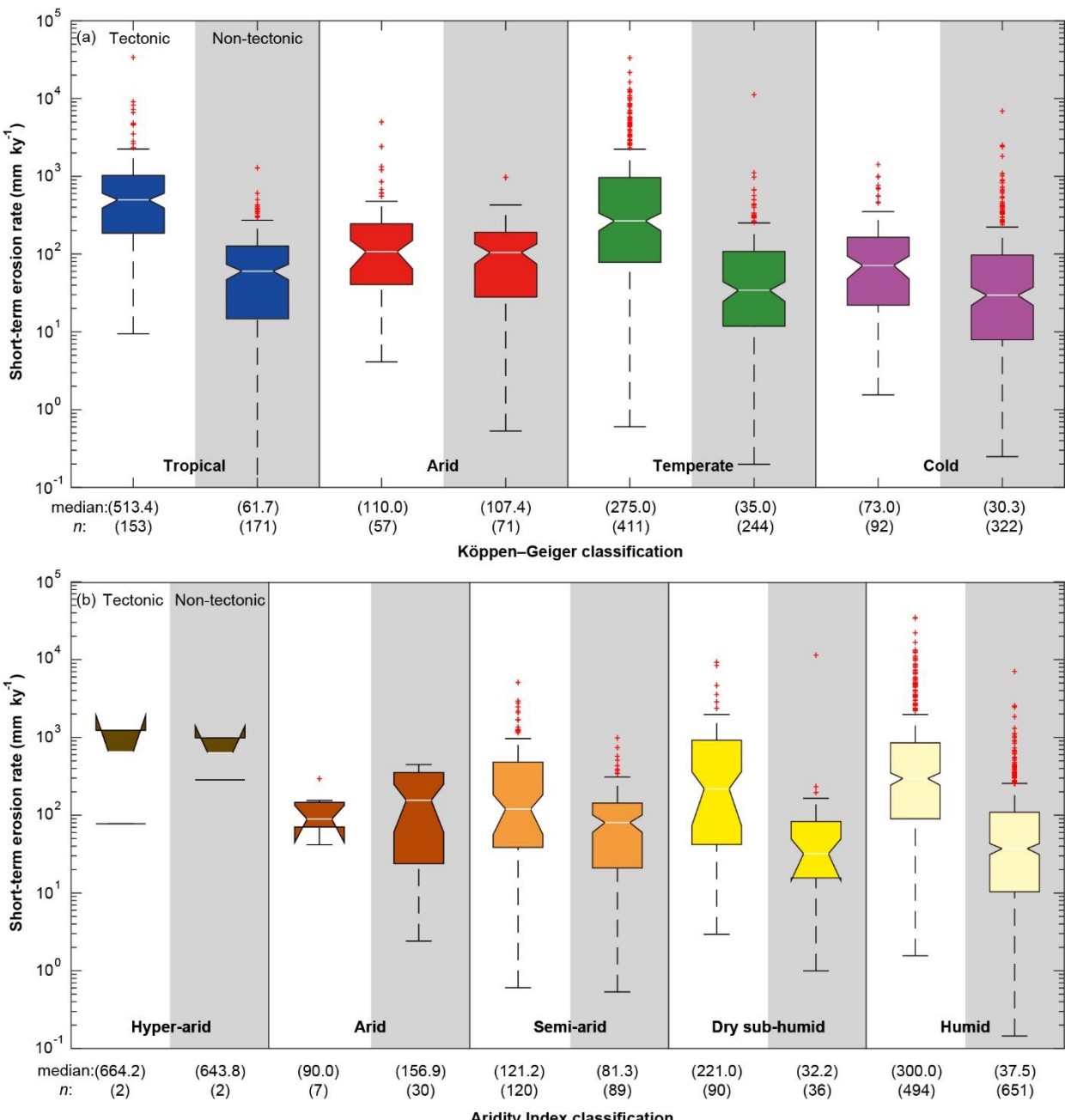

**Figure A3: Short-term erosion rates classified by tectonic (white backgrounds) and non-tectonic regions (grey backgrounds) for climate zones of Köppen–Geiger climate classification (a) and Aridity Index classification (b). Short-term erosion rates in tectonic regions are generally higher than in non-tectonic regions across all climate zones except for the Arid category of AI classification.**

**Author contribution.**

All authors contributed to the design, interpretation, and write-up of the study. S.-A.C. carried out the data compilation and data analyses.

**Competing interests.**

The authors declare that they have no conflict of interest.

**Acknowledgements**

K. M. acknowledges financial support from grant EP/T015462/1.

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
