# Peer review of "Exploring exogenous controls on short- versus long-term erosion rates globally"

_Earth Surface Dynamics, 2021_

## Referee Comment (RC1)

**Review for "Global analysis of short- versus long-term drainage basin erosion rates" by Chen et al.**

**General comments:**
This paper by Chen et al. is the latest attempt to gain a better understanding on the controls of climate and tectonics on erosion from the comparison of modern river loads with longer-term cosmogenic nuclide-derived denudation rates. The authors have my respect for attempting this comparison, as it is time-consuming to do. Nevertheless, I find the approach too general to be useful. For example, no attempt has been made to quantify uncertainties for short-term erosion rates, but knowledge of which should be crucial for when comparing two independent methods. Short-term rates are erosion rates from suspended sediments, while rates from cosmogenic nuclides are denudation rates, thus integrating over erosion and weathering. Strictly speaking, the two cannot even be compared. One way out would be to include dissolved river fluxes in short-term rates. While for rates from cosmogenic nuclides the authors rely on a previous compilation which has been carefully quality-checked, a quality check for the compilation of short-term rates is missing (see below).
As such, this study does not provide in my view significant scientific advances over previous studies that have carried out this comparison, and it is missing several substantial characteristics of a solid scientific manuscript (like uncertainty assessment). I am therefore against the publication of this manuscript at it is now.

**Specific/extended comments for each section**
**Intro:** What makes this study unique over the previous studies that compared these two different methods? (Besides, maybe, a larger dataset now available?). Why do we need yet another comparison? Is the comparison actually leading anywhere, as both methods have different biases… and the uncertainties associated might be too large to say anything beyond something that is better than a factor of 2 comparison? That alone could result in differences that are beyond the uncertainties.

**Methods:**
What does "compiled from published literature" mean for suspended sediments? Was there some initial quality check performed? For the USGS data, 2 criteria were used to confine the data (monitoring time and a basin area threshold). But, were there similar criteria for the other station data? Often, data is published were sediment rating curves are really poor, or monitoring times are really short. Especially in remote terrains, suspended sediment data is very sparse due to inaccessibility (in glacially impacted terrains) or due to infrequent rainfall and low discharge in general (dry regions). Hence, a rigorous data quality control and resulting means to use only the best data is needed first. Otherwise, any comparison can only be qualitative in nature and a quantitative comparison that even includes statistical analysis, as attempted by the authors, is useless. A useful endeavor for making short-term erosion rates better

comparable with cosmogenic nuclide denudation rates would be to associate an uncertainty to the former. Perhaps this could be done by MonteCarlo Simulation or so, but without having an uncertainty associated, the comparison remains qualitative. What does a factor of e.g. "1.4 higher" mean? Is this beyond uncertainty? As you may have guessed by now, in my view anything that this < factor of 2 between the two methods is actually a quite acceptable agreement. The problem is that not much more to be drawn if one of the methods does not have an uncertainty….

**Results:**
Another issue is that once datasets are compared to each other (short- vs. long-term rates), one should use the individual data from each basin/river only, meaning the data should be compared 1:1, i.e. only compare stations where there is actually short-term AND long-term data measured within an acceptable range of distance, or better even measured at the very same station). Only when trends with e.g. climate are analyzed for each short- or long-term dataset individually, the entire dataset might be used.

Section 3.4: This area-grouping makes sense and should have been done prior to the entire analysis. Otherwise, there is always the question of whether any trend observed may be due to the different number of observations within each bin….

Fig. 3: This trend found between the US-derived long-term erosion rates and MAP - is this trend also present in the entire dataset? If not, why is it present only in this dataset and how can then a global general interpretation be drawn if the global dataset does not show the same trend? (In line 376, the usage of 3,074 datapoints is mentioned in this regard. I´m confused, as in Fig 3, only the US data is used…Is the red line in Fig. 9 now using the entire dataset, or only a US-subset?)

Fig. 4: glacially impacted denudation rates higher than non-glacially impacted rates: That is nothing new. See reviews by Dixon et al. (2018) and Delunel et al. (2020, ESR) for the European Alps and the study by Ganti et al. 2016 (Sci Adv) that shows that cosmogenic denudation rates are likely affected by a time scale averaging bias. It´s a pity that these studies were not cited.

Fig. 5: I don´t think that an increase of 1.4 has any significance without analyzing uncertainties.

Fig. 6: What kind of figure is this? What does the bar legend indicate? Number of observations???

**Discussion:**
Section 4.1: A key point for the relation between long-term erosion and MAP is the LOWESS smoothing method. However, there is no reference nor any other further

information given how this smoothing works (averaging window?). Given that the resulting shape of the pattern is so much different than that found by others, I would encourage these authors to provide more information on it. See also my comments to Fig. 3 that are relevant here.

Section 4.2: What are the actual apparent ages (integration time scales) of the long-term data? Given that denudation rates are typically high (>0.5 mm/yr or so) in glacially- impacted regions, the resulting integration time scale are low (<1200 yrs), and do therefore not integrate over the last 25-15 ka. Same problem for Section 3.2.

Section 4.3: Same here as for Fig. 5.

Section 4.4: Sorry, I don´t get where this leads to. I find the section too general to be useful. Why make such a fuss about an absent relation between erosion and drainage area? Usually people use such an absent relation to show that there data is NOT influenced by sampling location… This section jumps from one topic to another without any clear red thread….The second para is ok for what the first-order observation is… (the fact that the larger the basin, the better the agreement between short- and long-term erosion rates). Last para: An $R^2$ value of 0.24 or 0.29 does not describe a significant relationship.

---

## Author Comment (AC1)

Dear Referee:

We thank you for the detailed and constructive review on our manuscript. We have considered all the comments and we will edit the manuscript in a revision where the description and explanation are unclear, misleading, or incomplete. First, we address the general comments, and then provide responses to specific comments below.

In our study, short-term erosion rates were estimated from suspended sediment yields, which likely underestimate basin-averaged erosion rates due to the lack of bedload and dissolved load data. Unfortunately, data of the latter two types of sediment transportation are too scarce, and the geographic distributions of these data are too uneven to meta-analyse them between climate zones at the global scale. Therefore, we did not include these data for the analyses. In addition, the environmental controls of bedload and dissolved load are complex and may show different characteristics from the suspended load. For example, the percentage of bedload to the total load tends to be higher in mountain regions and drylands (Dedkov and Mozzherin, 1996; Singer and Dunne, 2004), but the percentage of dissolved load seems to be higher in tropical regions and lower in drylands (Milliman and Farnsworth, 2013). Therefore, a systematic correction of short-term erosion rates is not possible. In order to enhance the reliability of short-term erosion rates, we suggest providing the globally-averaged percentage of bedload and dissolved load in the Methods. Previous studies estimated that the bedload typically accounts for < 10% of the total load (Milliman and Meade, 1983), and the average dissolved load accounts for around 20% globally (Milliman and Farnsworth, 2013). Thus, the short-term erosion rate maybe on average ~30% higher than what we presented in the study. Given that the short-term erosion rates, estimated only by suspended sediment yield, are higher than long-term rates in each climate zone (Fig. 2; except for the Cold K–G zone), the potential underestimation of short-term rates should not alter the conclusion of anthropogenic influences on short-term erosion in this study.

In terms of the quality checking short-term erosion rates, the data compiled from the USGS have been confirmed by the data source and labelled as 'Approved for publication: Processing and review completed'. The data collected from publications cannot be checked systematically and uncertainty ranges will be highly variable due to several reasons (Milliman and Farnsworth, 2013): the variety of measuring techniques over different periods of time, inadequate monitoring period (several rivers with historic records < 5 years, especially the shorter ones), watershed modification (resulted from dam construction or climate change), erroneous transcription of the data. Therefore, the quality of data is hard to assess and overcome, for which we will address the issue and limitation in the text. More generally, meta-analyses such as this one are inherently challenged by variability in the quality and substantiation of the underlying data (we will state this explicitly in the Methods). We have done our best to include data from the most reliable sources available (i.e. USGS, OCTOPUS).

In the reply below, the Referee's comments are in *black italics*, and our responses in blue.

Sincerely,

Shiuan-An Chen, Katerina Michaelides, Michael Singer and David Richards

**Intro:**

*What makes this study unique over the previous studies that compared these two different methods? (Besides, maybe, a larger dataset now available?). Why do we need yet another comparison? Is the comparison actually leading anywhere, as both methods have different biases… and the uncertainties associated might be too large to say anything beyond something that is better than a factor of 2 comparison? That alone could result in differences that are beyond the uncertainties.*

While some previous studies have either compared short- and long-term erosion rates between drainage basins or analysed erosion rates in particular regions (e.g. Kirchner et al., 2001; Portenga and Bierman, 2011; Wittmann et al., 2011), we are not aware of any studies comparing compiled short- and long-term erosion rates at the global scale. One of the key findings of our study is that a non-linear relationship exists between **long-term** erosion rates and climate, reflecting the balance between precipitation and vegetation cover (erosion rate peak in the semi-arid regions) which corroborates early theoretical work by Langbein and Schumm (1958) and Walling and Kleo (1979). However, we find that this relationship does not hold for short-term erosion rates as proposed by these former studies. This result was not mentioned by the reviewer but is a key finding and take-home message from the paper which directly addresses the reviewer's question as to what makes this study unique.

**Methods:**

*What does "compiled from published literature" mean for suspended sediments? Was there some initial quality check performed? For the USGS data, 2 criteria were used to confine the data (monitoring time and a basin area threshold). But, were there similar criteria for the other station data? Often, data is published were sediment rating curves are really poor, or monitoring times are really short. Especially in remote terrains, suspended sediment data is very sparse due to inaccessibility (in glacially impacted terrains) or due to infrequent rainfall and low discharge in general (dry regions). Hence, a rigorous data quality control and resulting means to use only the best data is needed first. Otherwise, any comparison can only be qualitative in nature and a quantitative comparison that even includes statistical analysis, as attempted by the authors, is useless. A useful endeavor for making short-term erosion rates better comparable with cosmogenic nuclide denudation rates would be to associate an uncertainty to the former. Perhaps this could be done by MonteCarlo Simulation or so, but without having an uncertainty associated, the comparison remains qualitative. What does a factor of e.g. "1.4 higher" mean? Is this beyond uncertainty? As you may have guessed by now, in my view anything that this < factor of 2 between the two methods is actually a quite acceptable agreement. The problem is that not much more to be drawn if one of the methods does not have an uncertainty….*

Erosion rate, suspended sediment yield, and sediment flux data were collected from published papers, textbooks, and the USGS website for estimating short-term erosion rates. The USGS data have been quality checked. However, there are difficulties in estimating the quality and uncertainty of data from other sources because the data were collected with various methods at different time periods. The two criteria used for USGS data were not used for collecting data from other sources. While the majority of suspended sediment yields from publications were recorded over 5 years, several records were shorter than this period of time. The reason for setting a basin-area threshold for USGS data is to enable comparison to similar basin sizes from which the long-term erosion rates were obtained (i.e. from the OCTOPUS database). The basin size of short-term erosion rates from other sources are generally larger (e.g. rivers included by Milliman and Farnsworth, 2013 include those discharging to the ocean, not the tributaries). We acknowledge the challenges in summarising suspended sediment data over a longer period of time, especially in cases where the rating curve between concentration and discharge shifts (we have published on this issue, Singer and Dunne, 2001). Nevertheless, we have relied on published sediment flux values from highly cited and well-regarded studies (Milliman and Farnsworth, 2013), which contain descriptions of data quality control. Therefore, we feel the conclusions from our meta-analysis of published short-term rates are robust. We will edit the explanation in the Methods.

**Results:**

*Another issue is that once datasets are compared to each other (short- vs. long-term rates), one should use the individual data from each basin/river only, meaning the data should be compared 1:1, i.e. only compare stations where there is actually short-term AND long-term data measured within an acceptable range of distance, or better even measured at the very same station). Only when trends with e.g. climate are analyzed for each short- or long-term dataset individually, the entire dataset might be used.*

We agree that it would be ideal to compare short- and long-term erosion rates at the same location. However, there are minimal data available ($n < 100$) that satisfy this criterion (Table 1 below). We also tried to extract long-term erosion rates within 200 km of all short-term rate points. Similarly, the number of paired data is generally less than 100 for each Aridity Index categories (Table 1 below). The comparison becomes meaningless if the distance threshold is further extended. Therefore, we chose to combine all data within each climate zone, assuming that with a large number of data, the variability within drainage basins will be reduced and some climatic signal on erosion rates may be revealed.

**Table 1: The number of paired data of short- and long-term erosion rates at the same location and within 200 km.**

| Number of paired data | Aridity Index categories | | | | | Sum |
|---|---|---|---|---|---|---|
| | Hyper-arid | Arid | Semi-arid | Dry sub-humid | Humid | |
| Same location | 1 | 5 | 6 | 15 | 52 | 79 |
| < 200 km | 2 | 12 | 76 | 53 | 439 | 582 |

Section 3.4: *This area-grouping makes sense and should have been done prior to the entire analysis. Otherwise, there is always the question of whether any trend observed may be due to the different number of observations within each bin....*

Below (Fig. 1) we present the erosion rates of basins smaller than 2,500 km$^2$. The patterns displayed in this figure are the same as the patterns using ALL data presented in the manuscript. One difference is that the short-term erosion rates are more variable between climate zones compared to the short-term rates using ALL data (main manuscript), which is likely a result from the limited short-term erosion data from smaller basins. Therefore, breaking down the data into even more area-group categories would substantially limit the number of data points within each climate zone.

[Figure]

Figure 1: Long- and short-term erosion rates of basins smaller than 2,500 km$^2$ for climate zones of Köppen–Geiger climate classification (a) and Aridity Index classification (b).

Fig. 3: *This trend found between the US-derived long-term erosion rates and MAP - is this trend also present in the entire dataset? If not, why is it present only in this dataset and how can then a global general interpretation be drawn if the global dataset does not show the same trend? (In line 376, the usage of 3,074 datapoints is mentioned in this regard. I´m confused, as in Fig 3, only the US data is used…Is the red line in Fig. 9 now using the entire dataset, or only a US-subset?)*

We only analysed the relationship between MAP and erosion rates in the USA because of the widespread systematic availability of MAP data across the USA – and mainly for comparability with other studies which tend to use MAP instead of Aridity Index. Nevertheless, the relationship between global long-term erosion rates and Aridity Index categories (Fig. 2b), show a similar result to the relationship with MAP across the US, with a peak in erosion rate within the Dry sub-humid zone. The only difference between these two results is that the Aridity Index classification cannot resolve erosion rates in the humid regions since all data are lumped into one bin. We will edit the legend of Fig. 9 to emphasise that the line is drawn from the USA data.

Fig. 4: *Glacially impacted denudation rates higher than non-glacially impacted rates: That is nothing new. See reviews by Dixon et al. (2018) and Delunel et al. (2020, ESR) for the European Alps and the study by Ganti et al. 2016 (Sci Adv) that shows that cosmogenic denudation rates are likely affected by a time scale averaging bias. It´s a pity that these studies were not cited.*

Thank you for the suggestion. We will add the literature in the Introduction and Discussion.

Fig. 5: *I don´t think that an increase of 1.4 has any significance without analyzing uncertainties.*

The magnitude of a difference between distributional medians does not indicate whether or not the difference is significant. Significance in this case is tested via the Kruskal–Wallis test, which showed that these two distributions are indeed statistically different (see manuscript). Regarding the issue of uncertainties in the underlying data, given that: 1) the short-term erosion rates in Fig. 5 are all generated with a consistent data collection procedure; and 2) the numbers of data points in agricultural and non-agricultural regions are extensive ($n$ = 826 and 732, respectively).

Fig. 6: *What kind of figure is this? What does the bar legend indicate? Number of observations???*

Figure 6 is a density scatter plot. The bar legend indicates the number of data points in each pixel. We will clarify this in the figure caption.

**Discussion:**
Section 4.1: *A key point for the relation between long-term erosion and MAP is the LOWESS smoothing method. However, there is no reference nor any other further information given how this smoothing works (averaging window?). Given that the resulting shape of the pattern is so much different than that found by*

*others, I would encourage these authors to provide more information on it. See also my comments to Fig. 3 that are relevant here.*

The LOWESS smoothing method uses locally weighted linear regression to smooth data, similar to the moving averaged method, which also smooth data locally by neighbouring data points, but the LOWESS uses linear polynomial regression rather than the average value (Cleveland, 1979). We drew the LOWESS regression by the built-in function, smooth, in Matlab, to highlight the pattern of erosion rates (Fig. 3a). We set the polynomial as "linear", the span as "30% of data points", and the robust as "off".

Section 4.2: *What are the actual apparent ages (integration time scales) of the long-term data? Given that denudation rates are typically high (>0.5 mm/yr or so) in glacially- impacted regions, the resulting integration time scale are low (<1200 yrs), and do therefore not integrate over the last 25-15 ka. Same problem for Section 3.2.*

According to Brown et al. (1995), the timescale of erosion rate of ~200 mm kyr$^{-1}$ (in past glaciated regions of the Temperate zone; Fig. 4) is around $10^4$ years, which may or may not have experienced the last ice age directly. However, the enhanced sediment production by former glacial processes can be transported by following fluvial processes during warmer periods, as shown by Ganti et al. (2016) (the erosion rates increase over shorter timescales from millions of years to decades). Since a significant difference in long-term erosion rates between past glaciated and non-glaciated regions is shown in our data, we believe that the erosion rates in mid- and high-latitude regions are, at least, indirectly enhanced by former glacial processes. However, we fully acknowledge that this is a subject of considerable debate in the literature, so we will add a fuller explanation in the Discussion.

Section 4.3: *Same here as for Fig. 5.*

Section 4.4: *Sorry, I don´t get where this leads to. I find the section too general to be useful. Why make such a fuss about an absent relation between erosion and drainage area? Usually people use such an absent relation to show that there data is NOT influenced by sampling location… This section jumps from one topic to another without any clear red thread….The second para is ok for what the first-order observation is… (the fact that the larger the basin, the better the agreement between short- and long-term erosion rates). Last para: An $R^2$ value of 0.24 or 0.29 does not describe a significant relationship.*

We feel it is necessary to explain the absent relationship between erosion rate and drainage area because the existing theory (i.e. stream power incision law) predicts a positive influence of drainage area on erosion rate, whilst some other studies present an inverse relationship (e.g. Milliman and Syvitski, 1992; Milliman and Farnsworth, 2013). Therefore, we proposed potential controls on the unclear relationship to highlight the complexity of drainage basin environments. We will add some clarification to the beginning of this section to explain why we are presenting this information. $R^2$ values do not "describe significance" of a relationship, but rather they quantify the proportion of the variance explained by the relationship. The significance of a

regression is measured by a $P$-value, and even relationships with very small $R^2$ values may be significant. The $P$-values of the relationships between long-term erosion rates and the slope gradient and total relief are both lower than 0.01 ($P$ = 4.23×10$^{-36}$ and 1.5×10$^{-32}$, respectively). We will add these values in the manuscript.

**Reference:**

Cleveland, W. S.: Robust locally weighted regression and smoothing scatterplots, Journal of the American Statistical Association, 74, 829–836, https://doi.org/10.1080/01621459.1979.10481038, 1979.

---

## Author Response (AR1)

Dear Referees:

We thank all three reviewers for the detailed and constructive reviews of our manuscript. We have considered all the comments and we have re-written most of the manuscript accounting for the reviews (Abstract, Introduction, Discussion, Conclusions). We made edits to the Methodology section to emphasise uncertainties in the data and to note all caveats and assumptions. . Below, we address the some recurrent comments first, and then provide point-by-point responses to specific comments. *Black italics* = reviewer comments, Blue = author response.

Sincerely,

Shiuan-An Chen, Katerina Michaelides, David Richards, and Michael Singer
* * *
**1. Suspended sediment, bedload and dissolved load**

In our study, short-term erosion rates were estimated from suspended sediment yields, which makes up the majority of sediment export from basins around the world. The omission of bedload and dissolved load data may underestimate basin-averaged erosion rates slightly but these data are too scarce, and too uneven to meta-analyse between climate zones at the global scale. A meaningful, systematic correction of short-term erosion rates is not possible due to variations in the controls on the type of sediment load between basins. For example, the percentage of bedload to the total load tends to be higher in mountain regions and drylands (Dedkov and Mozzherin, 1996; Singer and Dunne, 2004), but the percentage of dissolved load seems to be higher in tropical regions and lower in drylands (Milliman and Farnsworth, 2011). Previous studies estimated that the bedload typically accounts for < 10% of the total load (Milliman and Meade, 1983), and the average dissolved load is even less but with significant variation (Milliman and Farnsworth, 2011). For example, in some dryland basins, dissolved load is as low as ~ 0.2% (Alexandrov et al., 2009). Given that the short-term erosion rates, estimated only by suspended sediment yield, are higher than long-term rates in each climate zone (Fig. 2; except for the Cold K–G zone), the potential underestimation of short-term rates should not alter the conclusion of anthropogenic influences on short-term erosion in this study.

**2. Data quality checking**

The data compiled from the USGS have been confirmed by the data source and labelled as 'Approved for publication: Processing and review completed'. The data collected from published studies cannot be checked systematically and uncertainty ranges will be highly variable due to several reasons (Milliman and Farnsworth, 2011): variation in measuring techniques, short monitoring periods (several rivers with records < 5 years), watershed modifications (resulting from dam construction or climate change), and erroneous transcription of the data. Therefore, data quality is hard to assess and standardise and we now address these limitations in the text. More generally, meta-analyses such as ours are inherently challenged by variability in the quality and lack of standardization of the underlying data (we state this explicitly in the Methods). We have done our

best to include data from the most reliable sources available (i.e. USGS, OCTOPUS).

Despite differences in the biases and inherent assumptions in the calculations of long- and short-term erosion rates, and potential challenges in comparing the two, there is a precedence in the geomorphic literature for such comparisons, and numerous publications to demonstrate that this is generally acceptable. The following papers, cited in the manuscript, are examples of this precedence:

1. Bierman, P. R., Reuter, J. M., Pavich, M., Gellis, A. C., Caffee, M. W., and Larsen, J.: Using cosmogenic nuclides to contrast rates of erosion and sediment yield in a semi-arid, arroyo-dominated landscape, Rio Puerco Basin, New Mexico, Earth Surf. Process. Landf., 30, 935–953, 2005.
2. Clapp, E. M., Bierman, P. R., Schick, A. P., Lekach, J., Enzel, Y., and Caffee, M.: Sediment yield exceeds sediment production in arid region drainage basins, Geology, 28, 995–998, 2000.
3. Covault, J. A., Craddock, W. H., Romans, B. W., Fildani, A., and Gosai, M.: Spatial and temporal variations in landscape evolution: historic and longer-term sediment flux through global catchments, J. Geol., 121, 35–56, 2013.
4. Gellis, A. C., Pavich, M. J., Bierman, P. R., Clapp, E. M., Ellevein, A., and Aby, S.: Modern sediment yield compared to geologic rates of sediment production in a semi-arid basin, New Mexico: assessing the human impact, Earth Surf. Process. Landf., 29, 1359–1372, 2004.
5. Kirchner, J. W., Finkel, R. C., Riebe, C. S., Granger, D. E., Clayton, J. L., King, J. G., and Megahan, W. F.: Mountain erosion over 10 yr, 10 k.y., and 10 m.y. time scales, Geology, 29, 591–594, 2001.
6. Schaller, M., von Blanckenburg, F., Hovius, N., and Kubik, P.: Large-scale erosion rates from in situ-produced cosmogenic nuclides in European river sediments, Earth Planet. Sci. Lett., 188, 441–458, 2001.
7. Wittmann, H., von Blanckenburg, F., Maurice, L., Guyot, J.-L., Filizola, N., and Kubik, P. W.: Sediment production and delivery in the Amazon River basin quantified by in situ–produced cosmogenic nuclides and recent river loads, GSA Bull., 123, 934–950, 2011.

**Referee 1:**

**Intro:**
*What makes this study unique over the previous studies that compared these two different methods? (Besides, maybe, a larger dataset now available?). Why do we need yet another comparison? Is the comparison actually leading anywhere, as both methods have different biases… and the uncertainties associated might be too large to say anything beyond something that is better than a factor of 2 comparison? That alone could result in differences that are beyond the uncertainties.*

We hope our rewrite of the paper highlights the novelties of our analyses more clearly now. While some previous studies have either compared short- and long-term erosion rates within drainage basins or analysed erosion rates in particular regions (e.g. Kirchner et al., 2001; Portenga and Bierman, 2011; Wittmann et al., 2011), we are not aware of any studies comparing short- and long-term erosion rates at the global scale and exploring the various controls. One of the key findings of our study is that a relationship exists between **long-**

**term** erosion rates and climate, reflecting the balance between precipitation and vegetation cover (erosion rate peak in the dry sub-humid regions) which corroborates early theoretical work. However, we find that this relationship does not hold for short-term erosion rates as proposed by these former studies.

**Methods:**

*What does "compiled from published literature" mean for suspended sediments? Was there some initial quality check performed? For the USGS data, 2 criteria were used to confine the data (monitoring time and a basin area threshold). But, were there similar criteria for the other station data? Often, data is published were sediment rating curves are really poor, or monitoring times are really short. Especially in remote terrains, suspended sediment data is very sparse due to inaccessibility (in glacially impacted terrains) or due to infrequent rainfall and low discharge in general (dry regions). Hence, a rigorous data quality control and resulting means to use only the best data is needed first. Otherwise, any comparison can only be qualitative in nature and a quantitative comparison that even includes statistical analysis, as attempted by the authors, is useless. A useful endeavor for making short-term erosion rates better comparable with cosmogenic nuclide denudation rates would be to associate an uncertainty to the former. Perhaps this could be done by MonteCarlo Simulation or so, but without having an uncertainty associated, the comparison remains qualitative. What does a factor of e.g. "1.4 higher" mean? Is this beyond uncertainty? As you may have guessed by now, in my view anything that this < factor of 2 between the two methods is actually a quite acceptable agreement. The problem is that not much more to be drawn if one of the methods does not have an uncertainty….*

Erosion rate, suspended sediment yield, and sediment flux data were collected from published studies (listed in the supplemental data), and the USGS website for estimating short-term erosion rates. The USGS data have been quality checked. However, there are difficulties in estimating the quality and uncertainty of data from other sources because the data were collected with various methods at different time periods. The two criteria used for USGS data were not used for collecting data from other sources. While the majority of suspended sediment yields from publications were recorded over 5 years, several records were shorter than this period of time. The reason for setting a basin-area threshold for USGS data is to enable comparison to similar basin sizes from which the long-term erosion rates were obtained (i.e. from the OCTOPUS database). The basin sizes related to measurements of short-term erosion rates from other sources are generally larger (e.g. rivers included by Milliman and Farnsworth, 2011 include those discharging to the ocean, not the tributaries). We fully acknowledge the challenges in generalizing suspended sediment data over a longer period of time, especially in cases where the rating curve between concentration and discharge shifts (we have published on this issue, Singer and Dunne, 2001). Nevertheless, we have relied on published sediment flux values from highly cited and well-regarded studies (Milliman and Farnsworth, 2011), which contain descriptions of data quality control procedures. We have added an explanation in the Methods.

**Results:**

*Another issue is that once datasets are compared to each other (short- vs. long-term rates), one should use the individual data from each basin/river only, meaning the data should be compared 1:1, i.e. only compare*

*stations where there is actually short-term AND long-term data measured within an acceptable range of distance, or better even measured at the very same station). Only when trends with e.g. climate are analyzed for each short- or long-term dataset individually, the entire dataset might be used.*

We agree that it would be ideal to compare short- and long-term erosion rates at the same location. However, there are minimal data available ($n < 100$) that satisfy this criterion (Table 1 below). We also tried to extract long-term erosion rates within 200 km of all short-term rate points. Similarly, the number of paired data is generally less than 100 for each Aridity Index categories (Table 1 below). The comparison becomes meaningless if the distance threshold is further extended. Therefore, we chose to combine all data within each climate zone, assuming that with a large number of data, the variability within drainage basins will be reduced and some climatic signal on erosion rates may be revealed.

**Table 1: The number of paired data of short- and long-term erosion rates at the same location and within 200 km.**

| Number of paired data | Aridity Index categories | | | | | Sum |
|---|---|---|---|---|---|---|
| | Hyper-arid | Arid | Semi-arid | Dry sub-humid | Humid | |
| Same location | 1 | 5 | 6 | 15 | 52 | 79 |
| < 200 km | 2 | 12 | 76 | 53 | 439 | 582 |

Section 3.4: *This area-grouping makes sense and should have been done prior to the entire analysis. Otherwise, there is always the question of whether any trend observed may be due to the different number of observations within each bin….*

Below (Fig. 1) we present the erosion rates of basins smaller than 2,500 km$^2$. The patterns displayed in this figure are the same as the patterns using ALL data presented in the manuscript. One difference is that the short-term erosion rates in these smaller basins are more variable between climate zones compared to the short-term rates using ALL data (main manuscript), which is likely a result from the limited short-term erosion data from smaller basins. Therefore, breaking down the data into even more area-group categories would substantially limit the number of data points within each climate zone.

[Figure]

**Figure 1: Long- and short-term erosion rates of basins smaller than 2,500 km² for climate zones of Köppen–Geiger climate classification (a) and Aridity Index classification (b).**

Fig. 3: *This trend found between the US-derived long-term erosion rates and MAP - is this trend also present in the entire dataset? If not, why is it present only in this dataset and how can then a global general interpretation be drawn if the global dataset does not show the same trend? (In line 376, the usage of 3,074 datapoints is mentioned in this regard. I´m confused, as in Fig 3, only the US data is used…Is the red line in Fig. 9 now using the entire dataset, or only a US-subset?)*

Yes, the relationship between global long-term erosion rates and Aridity Index categories (Fig. 2b), show a similar result to the relationship with MAP across the USA (Fig. 3), with a peak in erosion rates in the Dry sub-humid zone. We only analysed the relationship between MAP and erosion rates in the USA because of the

widespread systematic availability of MAP (from gauges) data across the USA – and for comparability with other studies which tend to use MAP instead of Aridity Index.The only difference between these two results is that the Aridity Index classification cannot resolve erosion rates in the humid regions since all data are lumped into one bin. We have edited the legend of Fig. 9 to emphasise that the line is drawn from the USA data.

Fig. 4: *Glacially impacted denudation rates higher than non-glacially impacted rates: That is nothing new. See reviews by Dixon et al. (2018) and Delunel et al. (2020, ESR) for the European Alps and the study by Ganti et al. 2016 (Sci Adv) that shows that cosmogenic denudation rates are likely affected by a time scale averaging bias. It´s a pity that these studies were not cited.*

Thank you for these useful paper suggestions. We have added Delunel et al (2020) and Ganti et al. (2016) in the Introduction.

Fig. 5: *I don´t think that an increase of 1.4 has any significance without analyzing uncertainties.*

The magnitude of a difference between medians does not indicate whether or not the difference is significant. Significance in this case is tested via the Kruskal–Wallis test, which showed that these two distributions are indeed statistically different (see manuscript). Regarding the issue of uncertainties in the underlying data: 1) the short-term erosion rates in Fig. 5 are all generated with a consistent data collection procedures; and 2) the numbers of data points in agricultural and non-agricultural regions are extensive ($n$ = 826 and 732, respectively).

Fig. 6: *What kind of figure is this? What does the bar legend indicate? Number of observations???*

Figure 6 is a density scatter plot. The bar legend indicates the number of data points in each pixel. We have clarified this in the figure caption.

**Discussion:**
Section 4.1: *A key point for the relation between long-term erosion and MAP is the LOWESS smoothing method. However, there is no reference nor any other further information given how this smoothing works (averaging window?). Given that the resulting shape of the pattern is so much different than that found by others, I would encourage these authors to provide more information on it. See also my comments to Fig. 3 that are relevant here.*

The LOWESS smoothing method uses locally weighted linear regression to smooth data, similar to the moving averaged method, which also smooth data locally by neighbouring data points, but the LOWESS uses linear polynomial regression rather than the average value (Cleveland, 1979). We drew the LOWESS regression by the built-in function, smooth, in Matlab, to highlight the pattern of erosion rates (Fig. 3a). We set the

polynomial as "linear", the span as "30% of data points", and the robust as "off". We have added this explanation.

Section 4.2: *What are the actual apparent ages (integration time scales) of the long-term data? Given that denudation rates are typically high (>0.5 mm/yr or so) in glacially- impacted regions, the resulting integration time scale are low (<1200 yrs), and do therefore not integrate over the last 25-15 ka. Same problem for Section 3.2.*

According to Brown et al. (1995), the timescale of erosion rate of ~200 mm kyr$^{-1}$ (in formerly glaciated regions of the Temperate zone; Fig. 4) is around 10$^4$ years, which may or may not have experienced the last ice age directly. However, former glacial processes enhanced sediment production leading to higher transport rates by subsequent fluvial processes during warmer periods, as shown by Ganti et al. (2016) (the erosion rates increase over shorter timescales from millions of years to decades). Since a significant difference in long-term erosion rates between past glaciated and non-glaciated regions is shown in our data, we believe that the erosion rates in mid- and high-latitude regions are, at least, indirectly enhanced by former glacial processes. However, we fully acknowledge that this is a subject of considerable debate in the literature. We have added an explanation of the uncertainties and biases in the Methods.

Section 4.4: *Sorry, I don´t get where this leads to. I find the section too general to be useful. Why make such a fuss about an absent relation between erosion and drainage area? Usually people use such an absent relation to show that there data is NOT influenced by sampling location… This section jumps from one topic to another without any clear red thread….The second para is ok for what the first-order observation is… (the fact that the larger the basin, the better the agreement between short- and long-term erosion rates). Last para: An R$^2$ value of 0.24 or 0.29 does not describe a significant relationship.*

We have rewritten the Discussion and hopefully the narrative thread is clearer now. We feel it is necessary to explain the absent relationship between erosion rate and drainage area because existing theory (i.e. stream power incision law) predicts a positive influence of drainage area on erosion rate, whilst some other studies present an inverse relationship (e.g. Milliman and Syvitski, 1992; Milliman and Farnsworth, 2011). Therefore, we proposed potential controls on the unclear relationship to highlight the complexity of drainage basin environments. We have added the clarification to the beginning of this section to explain why we are presenting this information. $R^2$ values do not "describe significance" of a relationship, but rather they quantify the proportion of the variance explained by the relationship. The significance of a regression is measured by a $P$-value, and even relationships with very small $R^2$ values may be significant. The $P$-values of the relationships between long-term erosion rates and the slope gradient and total relief are both lower than 0.01 ($P = 4.23\times10^{-36}$ and $1.5\times10^{-32}$, respectively). We have added this value to the manuscript.

**Referee 2:**

*The manuscript by Shiuan-An Chen et al. describes a compilation of cosmogenically derived denudation rates and a compilation of short-term erosion rates from gauging data and couples these together to ask/answer questions about long- and short-term controls on erosion rates. I commend the grad student, first-author on a lot of work done in getting these datasets put together but I think the paper could be radically improved. I like the general idea of the paper but I found a couple of issues that I just can't get past in order to believe the results. First, denudation from cosmogenic nuclides is a combination of weathering products and erosion fluxes, but there is no mention of this in the paper or accounting for it by combining solute fluxes, or alternatively, convincing me that these data don't require this. Second, the paper reads like a choose-your-own topic where the authors pursue, in my opinion, too many different avenues. I would have preferred if they had decided, after doing all the background analyses that appear in the MS currently, which is the most interesting of the findings and then focusing the MS around that idea. Sadly(?), sometimes science involves doing work behind the scenes that doesn't need to appear in print anywhere. Personally, I find it really interesting that the analysis shows something very different from the Kemp et al., 2020 Nat. Comm. paper and would focus on that aspect given the strengths of the methods here and the flaws in that Kemp paper. Then, all the information/figures would be circling around this single topic. So I recommend a substantial revision to make the MS more readable/interesting to the reader. Currently, it has a PhD dissertation-chapter style, but not the style of a MS that tells a story or makes a compelling point through several lines of evidence.*

We have rewritten the manuscript, so hopefully our narrative thread is much clearer now. We acknowledge that we did not include the bedload and dissolved load in our estimation of short-term erosion rates due to the lack of data (see point 1 in main comments) and the difficulty of systematic correction (see point 2 in main comments), which we have explained in the Methods. However, given that the short-term erosion rates are significantly higher than long-term rates in all climate zones (except for the Cold K–G zone), the discrepancy between two methods should not alter the conclusion of anthropogenic influences on short-term erosion in this study. Kemp et al. (2020) showed anthropogenic influences on erosion rates at shorter timescales in North America that is consistent with our finding at the global scale.

Our aim is to meta-analyse erosion rates across timescales and identify their potential controls. Therefore, we feel it is necessary to include the key drivers of erosion (climate, glacial processes, anthropogenic activities, basin physiography) . We have changed our title to 'Exploring climatic and anthropogenic controls on short-versus long-term erosion rates globally' to better reflect the scope of this study.

**Abstract:**
*"Measuring erosion rates, analysing their temporal variations, and exploring environmental controls are crucial in the field of geomorphology because erosion through sediment transport in drainage basins shapes landforms and landscapes." <-- This is a confusing first sentence because it seems to imply that erosion is important but then that sediment transport is important. Erosion of sediment and transport of sediment are different things when viewed through the cosmogenic lens.*

We have completely rewritten the Abstract.

*"unpicking" <-- perhaps use 'unpacking' or better 'unraveling' instead? In any case, are they really controls or are they simply correlations? I'm guilty of doing this leaping myself so I know it is tempting.*

Thank you for the suggestion; we used "unraveling" instead. We start from the premise that the factors controlling erosion mechanistically are well-known, so all the factors we have chosen are known and well-established controls (not randomly chosen factors). The correlations allow us to examine the hierarchy of these controls for different timescales (i.e. long-term, short-term).

**Introduction:**

*I think the authors should take another try at creating a compelling abstract. This current abstract goes into details that are both irrelevant and either incorrect or imprecise. Fortunately, the authors don't need to correct each statement necessarily. It should be tightened up to focus on the paper and analysis done and conclusions drawn. I suggest writing papers backward: i.e. write the conclusions, then the discussion necessary to support them, then the results, methods, only parts of the intro necessary to tell the story at hand and then the abstract. This paper seems to be written the opposite way. It should really help rewrite the paper a bit more logically. Missing from the analysis are the other ways long and short-term data can be different including aliasing and the likehood of rare events (a la Kirchner, 2001 already cited) as well as the likely timescales of sediment storage and purging, which are relevant for sediment gauging data.*

*Here are some things that should be fixed and in some cases removed especially if irrelevant in a revision:*

*"The erosion rate of a drainage basin is an important geomorphic quantity because it reflects the net flux of sediment from source to sink in drainage basin and correspondingly, the rate and spatial pattern of landscape evolution." <-- There are a lot of concepts mashed together without regard for precise language or concepts here in this first sentence, which paints a negative first impression of the work, unfortunately.*

Thank you – we completely agree and we have rewritten the Abstract (and most of the rest of the manuscript) in response to these (and the other two reviewers' commnts).

*Line 50: I take issue with these statements. There are many indications that sediment storage can be on the order of millennia in floodplains. Also, it's unclear exactly what timescale they are thinking is negligible. There are very few cases where bedload and suspended sediment have both been quantified to a degree that one could actually say this. I would feel uneasy making this statement for the whole world from just a handful of studies.*

Thank you for this point. Please see response point 1 in general comments at the start of the rebuttal.

*Line 65: This whole paragraph is riddled with incorrect statements about the basin-wide cosmogenic method and missing are some critical parts that should be included. Cosmogenic nuclides don't assume no sediment storage. This is another example of imprecise language. If that was the case, no one would ever publish any cosmogenic nuclide data, because there's obviously sediment storage in watersheds. Cosmogenic nuclide basin-wide erosion rates do not assume no decay. It is often negligible but it fact we \*know\* that decay exists and is ongoing. Also, there is a lot of data to show that in many cases there \*are\* discrepancies with 10Be-derived erosion rate with differing grain size. In fact the earliest paper (Brown et al., 1995 that the authors cite) showed that this was the case. No erosion–deposition cycle? No, that's not how I understand it. Those papers don't say cyclicity doesn't exist anywhere that I remember. Quartz is not assumed to be equally present throughout the catchments - that is something that is checked, and if it isn't the case, it is corrected for or we don't use the method if it cannot be accounted for.*

Thank you for this – we agree and we have edited this paragraph to clarify the principles of cosmogenic nuclide-derived erosion rate.

*Cosmogenic nuclide analyses do actually need to assume that catchments have been receiving cosmic radiation throughout the entirety of the time they have been eroding the layer that has moved through the production zone and that the eroded sediment is coming from the surface. So, if a catchment has been glaciated - especially if only part of it has been glaciated - the concentration generally can't be used as reliable indicator of the erosion rate. Unless(!), the erosion rate was so high that the timescale of averaging is less than the glacier retreat age and it is only scraping off the very surface - not below the attenuation length. This is maybe the case in a couple places (New Zealand?) but it also creates circular reasoning given that the averaging timescale is determined from the erosion rate that might be too high owing to the glaciation itself. What a pickle! Another strategy is to not assume that t->infinity (in steady state) and to assign a deglaciation age to the "t" (time) in the erosion rate equation. Unfortunately, these conditions for "ok cosmo erosion rate data" from glaciated catchments are not met in the OCTOPUS database to the extent that you should trust them to make a definitive statement about glacial vs. non-glacial erosion rates. Any small difference you find in the two datasets could easily just be due to violation of the 10Be method. If the rates are lower for the glacial catchments, then you could maybe qualitatively infer something (I don't know how it would be quantified) since the glacial bias would cause the observed rates to be too high - not too low.*

Thank you for elaborating on this issue – it is indeed a pickle – and one that we cannot resolve in this paper. What we propose instead is to be very clear about all these caveats and uncertainties, but use the erosion data at face value. We are assuming that since researchers have made the decisions to measure cosmos in glaciated regions and have published these data to make statements about erosion rates of glaciated basins, then we can use the data in a compilation to make some tentative statements about erosion in these regions compared to other regions – again with very clear statements about the caveats.

*Line 80: The authors go from 'global' to 'nation' a little too quickly. (Also, I know this wasn't the intent but it sounds like the authors are saying the US is the only nation in the world.)*

This whole section has been rewritten.

Line 84: Typo: "of sediment yield" not "on sediment yield" but also saying non-linear here evokes the wrong concept. At mid-MAPs, the erosion rates are highest and are lower for very high MAPs and very low MAPs. Non-linear is a very vague way to say this. It is also non-exponential. You might say, there's a mid-MAP maximum or something like that. A similar fit is described much better in the later section that talks about the Mishra et al. paper.

We have edited the typo in the revised manuscript. We take your point about "non-linear" being a vague way to describe this relationship so we have edited the text to make it more specific.

Line 95: "Global analyses of short-term erosion rates from suspended sediment records suggest that a change to agricultural land cover has enhanced erosion rates by one to two orders of magnitude (Dedkov and Mozzherin, 1996; Montgomery, 2007; Wilkinson and McElroy, 2007; Kemp et al., 2020)." I don't think this is exactly right since these authors don't show a timeseries with agriculture imposed at some point. They simply show that agricultural rates are higher than different areas with other rates - sometimes at discrepant timescales, which is a fraught topic. Of the places currently in the literature (before this paper) where long term and short term rates are compared, in Covault et al 2013, which the authors cite, >50% of the long-term rates are higher than the short-term rates.

We agree that the language was imprecise – we have updated this to remove the suggestion of a timeseries.

Line 100: To my knowledge, nobody has definitively shown with data that vegetation actually plays a role like the authors are suggesting here (except perhaps Vanacker et al., 2007), it is a hypothesis.

We agree that the effect of vegetation on erosion may be hard to isolate, but many studies have demonstrated the link both with field data and modelling (e.g. Thornes, 1990; Collins and Bras, 2008; 2010).

Line 110: Similar comment to that above: 'higher rates are *associated* with glaciated terranes' might be a defendable statement (but not using cosmogenic nuclides, since they actually don't show this.) Are the authors actually saying that the decreased infiltration that fires create in soil surface geochemistry (that last for only ~1 week to months at the most) are responsible for higher long-term rates of erosion? There's really no good way (with cosmogenic nuclide data) to show which areas are burned more or less over the millennia of averaging.

According to previous studies, wildfires destroy vegetation cover and alter the hydrologic characteristics of the hillslope, leading to higher possibility of debris flow and landslides (Cannon et al., 1998; Meyer et al., 2001; Pierce et al., 2004). Although the time periods of these alterations are short compared to the time scale of cosmogenic nuclide data, these events provide large amounts of sediment flux over a short time

span, which is hypothesised to increase long-term 'averaged' erosion rate (e.g. Kirchner et al., 2001). We do not explicitly identify regions that are more susceptible to wildfires but support the idea that higher long-term rates may result from wildfires and other events with large magnitude and low frequency.

*Line 136: topology? I don't know that this is actually achieved.*

We have changed the term to "key topographic indicators" in the revised manuscript.

*Line 210: "To extract river profiles from the database for comparing topographic parameters with erosion rates, we chose a subjective distance threshold as 150 m between river profiles and erosion rate sampling points (i.e. selecting river profiles which are within 150 m to the closest erosion rate points), and calculated the mean slope gradient and total relief of river longitudinal profiles." I don't see why such a large window was used here. Usually, people sample for 10Be right on the river itself and this highlights a potential problem with the geoid or projection used, if the points are this far away from the actual river on the map. The OCTOPUS sites have made sure that the sites are approximately on river sites.*

We set this distance threshold is because the GLoPro database contains river channels extracted from STRM–DEM (with 30 m resolution) whist the erosion rates were sampled right on the rivers. Therefore, there may be discrepancy in the locations between these databases. Considering the resolution of STRM–DEM and channel width, we believe this threshold is reasonable for the analysis.

*Line 225: To a reader not familiar with kruskalwallis, could you describe the method and why it was chosen? This is fairly important because so many variables here interact so the authors could be/are conditioning on a collider. I think the kruskalwallis function does not help to eliminate this issue. I hope another reviewer covers some suggestions for stats that would be helpful to the author because I don't know what to use with so much nonlinearity and dependencies in the data. For example, higher precipitation and lower temps are actually \*created\* by mountain ranges. So how would one disentangle slope and elevation from those climatic effects. This is not the way to account for that.*

The Kruskal–Wallis is a nonparametric hypothesis test that compares the variance between multiple samples to determine whether they are from the same distribution. The Kruskal–Wallis test is suitable for determining whether two sets of erosion rates (with different environmental influences) are significantly different, where the underlying distributions may not be normal. The purpose here is to assist in determining differences in the data but not to determine complex relationships between environmental controls.

**Discussion and conclusions:**

*I assume these will radically change in scope/focus with submission of a major overhaul revision.*

Thank you – yes, we have completely rewritten the Abstract, Introduction, Discussion and Conclusions.

**John Jansen:**

**General comments**

*Chen et al. present a global meta-analysis of previously published data on denudation (erosion) rates from a wide selection of drainage basins. The data draw on two approaches to quantifying basin-scale erosion: suspended sediment (SS) yield and cosmogenic Be10 measured in fluvial sediments. Key findings are that short- and long-term erosion rates tend to differ significantly, and that this disparity is largely due to i) human activities, and ii) climate-related factors involving the role of plants, glacial history, topography, and scale effects associated with sediment storage in fluvial systems.*

*How timescale affects the quantification of erosion rates is a primary question in geomorphology that has received plenty of attention. Making use of large datasets to explore the problem is not novel, but to my knowledge the suspended sediment (SS) yield data has not been previously examined specifically in this way.*

*In my view the study would gain from some restructuring that clarifies the rationale behind each step of the analysis and leads to a more logical unfolding. At present the MS gives a jumbled impression, some passages are difficult to follow and several loose threads detract from the main arguments. The conclusions are mostly compatible with previous work, but my main concern is with a number of oversights that seriously weaken the standing of the work.*

Thank you for this point and we agree - we have re-written the entire manuscript to enhance the narrative.

*I have 5 main points that require some consideration (my other comments are keyed to line #).*

*(1) Treating Be10-derived erosion rates as long-term in comparison to the SS records is a valid approach, but it's also important to note that the Be10 integration timescale is a function of erosion rate. The integration timescale is conventionally calculated at one absorption depth scale ~ the time taken to remove ~ 0.6 m of rock under long-term steady erosion. This means Be10 integrates a continuum of timescales spanning 3 order of magnitude ($10^3$ to $10^6$ y), and this may have implications for how the erosion rates are interpreted, as noted below in my comments on Fig. 3a.*

We have clarified the timescales of integration in the revised manuscript.

*(2) There is a striking omission here of the time-dependence of the distribution of hiatuses in the sedimentary record known as the Sadler Effect. This issue lies at the heart of comparisons of erosion and deposition rates over different timescales and cannot simply be passed over without comment. There is a stack of recent papers, but Schumer & Jerolmack (2009, JGR) would be a good start.*

Sadler (1981) showed a systematic higher accumulation rate with shorter timescales, mainly resulting from the incompleteness of stratigraphic sections which is caused by the episodic sedimentation process. However,

this dependency on timescale is more apparent in the net accumulation environment at a single location (Sadler and Jerolmack, 2014). Since our compiled erosion rates were estimated from suspended sediment flux and $^{10}$Be concentration in fluvial sediments with various basin areas, rather than from lake deposits or floodplain stratigraphic records with small contributing areas, our result should not bias with timescale systematically.

*(3) In order to reflect short-term denudation, the SS data should be limited to specific sediment yields within the upland source zone only. In the transfer zone, downstream SS load is chiefly driven by sediment exchange between floodplains and channels: a function of sediment availability, not denudation (see Dunn et al. 1998, GSAB). If on the other hand, direct efflux from agricultural lands and plantation forestry is the driver then the authors need to make their case accordingly.*

The purpose of this study is to analyse basin-averaged erosion rate, which integrates the denudation of the hillslope, erosion and sediment transport across the entire basin, and identify the environmental controls on these processes. Suppose the data are limited to upland regions. In that case, the analysed basin areas will all be small, and the results may be biased.

*(4) I am not convinced the median is a legitimate choice of parameter for comparing datasets that overlap across several orders of magnitude. In such variable data, the median has no specific value other than being roughly in the middle. I would like to see a justification for the use of the median as the basis for comparing relative erosion rates.*

At the global scale, the characteristics of basins vary widely, leading to a large range of erosion rates between basins even after classifying the rates by environmental factors. In order to present the overall characteristics of erosion rates, we used the median value since it is less biased by extreme values (compared to the mean, for instance). Additionally, we conducted statistical tests to determine whether two sets of data are significantly different in terms of their medians and their distributions.

*(5) The topographic indices do not include mean hillslope gradient despite it being the strongest of all parameters tested by Portenga and Bierman (2011). The local river slope used here has practically no bearing on Be10-derived basin-scale erosion rate and there is no reason to expect it would either. Channel relief is not an effective substitute. The physical basis of the alternative approach presented here needs to be justified.*

The mean channel gradient and total relief of selected river channels were calculated from the entire channels (from the highest to the lowest points of the channel), not from the local-scale data. Although the data do not include hillslope topography, it should represent the topographic characteristics of the whole basin.

**Specific comments** [keyed to line #]

**Intro**

*35- Perhaps add that SS flux records also reflect the availability of fine sediment, and to some extent the production of fines via weathering and transport; e.g. some lithologies like NZ greywackes break down remarkably fast.*

We have rewritten this section to clarify.

*36- '…basin-averaged exposure ages', is not the correct phrase. As used here, Be10 abundances are modelled to yield surface erosion rates, not exposure ages, which generally impose a zero-erosion constraint—it's a different equation. Please correct this error throughout the MS.*

We have removed "exposure age" throughout the manuscript.

*52-3. This assumption of uniform bulk density seems reasonable for the purposes here but should probably be propagated through the uncertainty analysis. I didn't find any evidence of that.*

We acknowledge that the assumption of uniform soil bulk density is a limitation of estimating short-term erosion rates. Unfortunately, because of the lack of bulk density data around the basins where sediment yields were acquired, it is not possible to provide an uncertainty estimate of the density.

*63- The integration timescale of cosmogenic nuclides varies with erosion rate.*

We have included this explanation.

*68-73. I appreciate seeing an explicit statement of the method assumptions; however, I can suggest a few amendments. Sediment storage is not strictly a problem (reworking can be). Most important is that the sampled grains have been subject to long term steady erosion and continuous exposure to cosmic rays. Those two assumptions are violated by abrupt and deep erosion (e.g., landsliding), or long-term burial followed by erosion. Landscape-scale equilibrium is not really necessary (see Willenbring et al. 2013, Geol); if it were, practically all of the OCTOPUS dataset would be invalid. The key is that the erosional processes acting are more or less steady and there is a minimum of deeply shielded grains (e.g., from landsliding) in the sample collected. Please clarify the meaning of 'no erosion-deposition cycle'.*

Thank you for the suggestion. We have rewritten the methods to include a clarification of assumptions and uncertainties.

*74-76. See note above on the integration timescale of Be10. This statement is valid only for the longer-lived nuclides such as Cl36, Al26 and Be10 (not C14). It is not clear what is meant by 'stochastic events', but*

*perturbations such as landsliding and extreme floods that erode old sediment storages can potentially affect Be10 abundances, and certainly will affect abundances of in situ C14.*

We have clarified the types of cosmogenic nuclides mentioned here. The 'stochastic events' refers to episodic erosion with shallow depth.

*107- '…stripping of rock underneath basal ice' is not how it is usually described. Perhaps rephrase to something like 'Glaciers erode bedrock via quarrying and abrasion wherever subglacial conditions allow basal sliding.' The legacy of deep and steep walled glacial troughs prone to mass failure is another key reason why glaciated landscapes yield high sediment load.*

The Introduction has been rewritten.

*129- True, but deconvolving the effects of tectonics and climate is not really one of the aims of this MS. Perhaps confine the scope of this literature survey to the issues that are specifically addressed in the MS.*

Deconvolving the effects of tectonics and climate is not the aim of this study; however, we feel it is necessary to explain why both factors influence erosion rate and why it is hard to isolate one from the other.

*134- … glacial and periglacial processes.*

Thanks for the suggestion. We have included periglacial process throughout the manuscript.

**Methods**

*147- Was the Ray and Adam (2001) study used here because it classifies vegetation distributions at the LGM? Are there more up to date alternatives?*

Yes, we compiled the data from Ray and Adam (2001) because they provided the vegetation distributions at the LGM, representing the glacial and periglacial extents, which is relevant for our purpose.

*164- '…published literature' seems a bit general—perhaps refer to the Supp data here.*

Good point – we now refer to the Supplementary data where we mention published studies used in the paper.

*166-67. One of the strengths of the OCTOPUS dataset is that the nuclide data are recalculated from scratch with uniform methods and propagation of uncertainties. Has the same been done with the SS data including uncertainty analysis? e.g., the bulk density assumption? It would be good to see an effort in favour of reproducibility.*

See point 2 in the main comments above.

*182- It is a bit unclear whether the original K-G zones or the new modified versions were used. Fig. 4 is not clear to me. What is the purple representing? I cannot determine clearly where the previously 'glacial and proglacial zones' are exactly. Is the map indicating that the Tibetan Plateau was ice covered during the LGM? That idea has long been discredited (e.g. Heyman 2014, QSR), giving the impression that these glacial extents are a bit outdated.*

The K–G classification includes five main climate zones and 29 sub-zones, but we exclude the data from the Polar zone and only analysed data based on the rest of the four main zones. In Fig. 4, the glacial regions are represented by 'translucent' blue tone, so the purple areas are the overlaps of glacial regions at the LGM and the Cold K–G zone at present. We have edited the legend to clarify the tone. The glacial regions represented here are not only defined by ice coverage, but also tundra, and Polar and alpine desert. According to Ray and Adams (2001), the Tibetan Plateau was mainly covered by alpine desert, which is subjected to periglacial activity.

*188-194. Some repetition could be cut here.*

Done – rewritten most of the manuscript.

*191- In addition to the regions directly covered by Plio-Pleistocene glaciers, it is worth considering the widespread distribution of frost weathering associated with periglacial activity. At high latitudes, this is a far more extensive and more persistent control on sediment production and transport than ice sheets per se. The direct affects of glaciation extend far beyond those regions directly covered by Plio-Pleistocene glaciers. The great northern ice sheets fed prodigious amounts of sediment into surrounding glaciofluvial landscapes, which were in turn surrounded by a vast periglacial domain. Even today, about 18% of the ice-free terrestrial surface has a mean annual temperature <0°C and of course this expanded greatly during the LGM.*

We agree that both glacier and periglacial activity lead to higher erosion rates. In fact, the extent of glaciation we determined at the LGM includes areas broader than the ice coverage (e.g. tundra). We have edited the text for clarification.

*196- Perhaps 'glacial-interglacial cycles' rather than 'ice ages' which is a bit general; 'last ice age' presumably means the coldest part of the last glacial cycle, the global LGM (~27–19 ka). This is important in the context of the Be10 integration timescale noted elsewhere.*

We have edited these sentences.

*207-11. Owing to its scale dependence, estimates of 'mean slope gradient' i.e., hillslope gradient, have created some difficulties in previous meta-analyses (e.g., Willenbring et al. 2014, ESurf). I suggest you be specific about how this is calculated; expand on Chen et al. (2019). A bit puzzling. The topographic data*

*presented here include river profile concavity but not river slope, or some normalised version of steepness such as ksn. That seems a bit odd given that river slope is one of the main drivers of fluvial incision (via bed shear stress). Is this saying that it was local river slope that was measured within 150 m of the Be10 samples, not mean catchment hillslope gradient? Local river slope has practically no bearing on a Be10-derived erosion rate. In any case, I suggest the term 'reach-scale channel slope' be used instead of 'mean slope gradient'.*

Apologies for the confusion – we mean the river gradient (not the hillslope). The data recorded in the GLoPro database include the elevation and flow distance of each pixel throughout the entire rivers extracted from the SRTM–DEM. The mean gradient of selected river channels was calculated by dividing the total relief between the highest and lowest elevation along the 'entire channel' by the river length. Therefore, it is not the local channel slope. The 150 m threshold is the distance between the erosion rate sampling point and the nearest river channel selected for calculating the mean slope and total relief. NCI (normalized concavity index) is used instead of Ksn because it circumvents issues with stream power incision theory assumptions that channel erosion is intrinsically tied to an assumed relationship between river discharge and drainage area.

*226- Please expand on the description of the Kruskal-Wallis test. I understand the K-W is an old school ANOVA based method, but it needs to be justified here. Why is K-W the most appropriate tool to use among so many others?*

We chose the K–W test is because it is a nonparametric hypothesis test that compares the median values of multiple samples to determine whether they are from the same distribution. Since we analysed the tendency of erosion rates between environmental controls by the median values and the data may not be normally distributed, we consider K–W test is suitable for determining whether two sets of erosion rates are significantly different. We have added the explanation of K–W test. See also our response to Reviewer 2 about this point.

**Results**

*Fig. 2 is a nice and clear representation of the data. My first impression was that all these erosion rates overlap at the interquartile range and so demonstrate remarkable similarity despite spanning such different timescales! Fig. 2 shows that most sample sets span an order of magnitude in the interquartile range and 2–4 orders for the 5–95 percentile (I presume the whiskers are 5–95, though I could not find it stated anywhere). I have several concerns.*

We agree that erosion rates overlap between climate zones and different timescales which is a result of various environmental controls across the globe. The whiskers are not the 5th and 95th percentiles of data. We drew the boxplots with the default setting of the MATLAB, which defines the whiskers as between $q_1 - 1.5 \times (q_3 - q_1)$ and $q_3 + 1.5 \times (q_3 - q_1)$; where the $q_1$ is 25th percentile whilst the $q_3$ is 75th percentile.

*(i) How were those outliers defined? Is there a physical basis for excluding those data? In my view a solely statistical reasoning for defining the outliers is not justifiable because these are simply descriptive classes; they do not imply a model distribution that would dictate the shape of the tail, for instance. Excluding outliers implies that there is a problem with those data. But what would that problem be other than they are in the tail of the distribution? This is important because excluding data naturally affects the shape of the distributions—possibly a lot (the Portenga & Bierman 2011 study does the same).*

The outliers shown in the boxplots were defined by the equation mentioned above (from the default setting of MATLAB). However, these data were not excluded in our analyses. We used ALL of the data to calculate the median values and conduct the hypothesis test.

*Given that the outliers are all at the upper end of the data range, I would guess that the distributions are close to log normal. If so, then it probably also means they could benefit from log-transformation before plotting. Out of curiosity, I plotted the SS data provided in the Supplement using violin plots rather than the standard box-whisker. The violins have obvious advantages and I suggest the authors try these out.*

Thanks for the suggestion. We tried using a log-scale on the boxplots to show the distributions of lower erosion rates more clearly but the results do not look as straightforward as in the original boxplots.

*(ii) If one includes the outliers for a moment, most of the classes span >4 orders of magnitude in erosion rate, which suggests a severely undetermined problem. In my view these climate classes are simply not discriminating enough—there are too many other factors at play.*

We agree that many other environmental factors influence the erosion rates, such as tectonics and lithology, which were not analysed in this study and may be the causes of large range of erosion rates. However, we aimed to determine broad differences between climate categories and across timescales. Statistical tests confirm these differences suggesting that the climate is a main control that is evident from the data.

*(iii) I find it difficult to understand why the median is a legitimate choice of parameter for comparing datasets that overlap across several orders of magnitude. In such distributions, which are potentially polymodal, the median has no specific value other than being roughly in the middle. Further, the median ignores the magnitude-frequency of events that drive sediment yield. For instance, a good proportion of the sediment yield in the tropics is the result of tropical cyclones and another large fraction is related to seasonal burning which has been practiced by indigenous people over ~$10^4$ timescales. My guess is that the median would fall in between those two. Is that a good model? In my view, the authors need to work a bit harder to convince the reader here and especially argue why the K-W is the best and most appropriate tool to use.*

Each statistic and hypothesis test has its limitation. In our opinion, for data widely distributed, the median value is suitable to represent the overall trend since it is not biased by the extreme values and the type of data distribution. The K–W is also appropriate to justify whether the median values are significantly different.

*Table 1. Tables are never a nice way to present an argument. To clarify, the 'erosion rates between climate zones' are merely the median erosion rates, not the full distributions. Is that correct?*

The purpose of this table is merely to show the significance of differences in the data in Fig. 2. Although the K–W test is based on the median values, the *P*-value is influenced by the number and ranks of all data.

*Fig. 3. Interesting plot; I like the LOWESS approach. Where are the uncertainties on the erosion rates?*

We have added the uncertainty range based on the long-term erosion rate errors reported in the OCTOPUS database.

*Fig. 3a. Combining present-day precip rates with Be10-derived erosion rates, which integrate a range of timescales, might have some implications worth considering. Two related issues that come to mind with regard to the bump in the data at MAP <1000 mm: the higher erosion rates (~1000 mm/kyr) are integrated over a timeframe of ~600 y while the slower rates (~20 mm/kyr) are integrated over a timeframe of ~30 kyr. It is well established that presently arid parts of the American West experienced much wetter conditions over the transition from full glacial to the present-day interglacial (REF), and this is the same timescale spanned by the Be10 samples. The changes in MAP are sufficient to move these data to the right (possibly forming a cluster alongside the 'humid trough shown now). I'm just speculating here ...*

We totally agree with this comment. But in the absence of systematic, global, paleoprecipitation data, we have to make assumptions about climate, and using present-day MAP is the only viable option. All other studies that investigate [10]Be-derived erosion data against MAP, make the same assumptions.

*274-275. This is good idea to exploit the region between the former maximum ice margins and the non-glaciated temperate zone but this is not well executed, in my view. A couple of things require clarification given that the glaciated and non-glaciated zones seem to overlap to a great extent:*

*(i) What is the purple zone in Fig. 4?*

The glacial regions at the LGM are represented by 'translucent' blue tone, so the purple areas are the overlaps of glacial regions at the LGM and the Cold K–G zone at present.

*(ii) Note that the LGM ice limits were visited only very briefly; ice cover for most of the Pleistocene was a small fraction of the LGM max. In other words, how useful is the LGM limit as an index of glacial erosion given that most of the glacial forefield is mantled with drift in places hundred of metres thick.*

The purpose of this analysis is to confirm whether mid- and high latitude regions subjected to glacial and periglacial processes at colder periods are associated with higher long-term erosion rates. Therefore, our

finding is not limited to the LGM and directly under ice coverage, but where the land surface was influenced by glacial and periglacial processes when the temperature was colder than the present.

*(iii) Not all glaciers are alike. Polar ice masses erode slowly (<10-100 mm/kyr) owing to their frozen-beds. It is true that the ice sheets complicate the quantification of large-scale erosion rates, but short-term rates linked specifically to glaciers have been quantified (see Hallet et al., 1996, GPC; Koppes & Montgomery 2009, Nat.Geo).*

In our compiled data, the Polar K–G zone was excluded. Therefore, the regions covered by (steady) ice sheets with low erosion rates should be limited in our targeted areas.

*277- I don't think it is reasonable to characterise this comparison as 5-fold discrepancy without mentioning that it is merely the medians being compared (as stated in the Fig. 4 caption). Most of the data (5–95) overlap across 2 orders of magnitude.*

We have clarified this in the text.

*279- As noted above, this effect is not just driven by glaciers but the full range of cold climate processes.*

Thanks for the clarification. We have added the related processes (e.g. periglacial activity) in the text.

*290- Puzzling why so little difference! It suggests the classes are not effective. Can this be improved somehow—there must be a stack of global datasets describing anthrogenic activities nowadays.*

We acknowledge that the difference is smaller compared to other studies. We speculate that one of the main reasons for this discrepancy is that here we may be underestimating the amount of area that is influenced by anthropogenic activity, based on our defined threshold of > 50% agricultural area. Another possibility is that our analysis may be including more short-term erosion rates sampled in anthropogenically impacted regions, where substantial soil and water conservation efforts in upstream basins, as well as engineering structures (e.g. dams) that trap sediment may result in artificially lower sediment yields. However, the short-term erosion rates are not significantly different between the two agricultural regions, whilst the difference between anthropogenic and non-anthropogenic regions is significantly different, showing that human activity is associated with higher short-term erosion rates.

*304-305. Finding that $R_{S/L} > 1$ is a predicted outcome of the time-dependence observed in the sedimentary record known as the Sadler Effect.*

The Sadler Effect describes the timescale dependency on erosion rate estimated from the net accumulation environment. Since our data were calculated from sediment yields of gauging stations and [10]Be concentration

in fluvial sediments, this effect should not be the main cause of $R_{S/L} > 1$. See also our response to your point 2 above.

*Fig. 7. Aside from the issue that these histograms obscure the enormous spread of these data, I'm interested to understand what causes the differential between the tropical and arid datasets. Could these trends be the magnitude-frequency factors emerging? For instance, the flood frequency curves in the tropics are characteristically flat whereas those in arid regions are steep.*

We agree that the likely controls are differences in hydrological regimes. This is now explicityly discussed in the Discussion (lines 515-530).

*Fig. 8. Not well conceived in my view. I cannot grasp what the right panel is aiming to show. In (c) no river slopes exceed about 0.05, unlike in (a), because SS load derives mainly from reworking of fine sediment fills found predominantly in the transfer and sink zone at low channel slopes. At higher slopes there is little availability of SS load, so I expect that this plot is largely reflecting landscapes with argillaceous lithologies or glacial settings–not an effective way of exploring how slope affects erosion rates.*

The right panel shows that the topographic controls on short-term erosion rates are not as apparent as for long-term rates. The data shown in Fig. 8c, d are distributed across the globe (Fig. 2 below), which should reduce biases in basin and river characteristics of, yet the long-term and short-term datasets may be derived from different locations (hence slightly different slopes). However, there is still plenty of overlap in the x-axis, so it is a reasonable comparison. We have clarified the potential reasons for the lack of signal in short-term erosion rates within the Discussion.

[Figure]

**Figure 2: Locations of short-term erosion rate data used for the topographic analyses.**

**Discussion**

*336- Given that this study is based upon comparing short- and long-term erosion rates, this part of the Discussion ought to include some consideration of the effects of the time-dependence observed in the sedimentary record known as the Sadler Effect. I suggest a brief outlining of how it has been recognised by*

*previous workers, followed by some analysis demonstrating that the observed findings of $R_{S/L} > 1$ are not a simple outcome of the Sadler Effect.*

*It is my understanding that Be10-derived basin-scale erosion rates are not subject to this bias for the reason that they incorporate the erosional-depositional dynamics across a wide range of ground surfaces in the basin, some eroding some not, and this effectively neutralises the time-dependence. Whether or not the SS load data reported here is subject to a time-dependent bias is for the authors to demonstrate.*

The Sadler Effect describes the average accumulation rate, which decreases with a longer timescale. Previous studies hypothesise that it may result from post-depositional compaction or long-term evolution of geomorphic system (e.g. crustal subsidence); however, the primary cause should be discontinuous sedimentation (Sadler, 1981). Therefore, erosion rate estimated from the suspended sediment yield at gauging stations does not apply to this case. In terms of [10]Be-derived erosion rate, as the reviewer explains, the data include both erosion and deposition processes across various basin areas, so the influence of Sadler Effect should be minimal (Sadler and Jerolmack, 2014). See response to your point 2 above.

*346- This could be rephrased to be less general. As it reads now, this conveys very little information and reports findings that more or less echo those of previous work. A major part of the analysis is not global, it is restricted to the USA.*

We have rewritten all the Discussion. Although the influence of MAP on long-term erosion rates was analysed from the USA data, a similar trend was found between aridity (AI) and long-term erosion rates at the global scale (Fig 2b).

*352- Perhaps a bit too generalised. The non-linearity in this relationship is most likely a function of the response of plants (soils reinforced by roots, rainfall interception, weathering etc.).*

We agree that this relationship is largely caused by the plants (although the plants are also influenced by the climate). However, we have included a detailed explanation in the Introduction, so we did not repeat it in the Discussion.

*362-64. The question is, why does Mishra et al. (2019) differ from the curve produced here? Perhaps the authors could raise some explanations here.*

Good point! Their data scatter looks very similar to ours with a peak in the ~600mm region. However, due their choice of trendline fit (3[rd] order polynomial) they obtain a different curve which shifts the peak to the right. We actually think that their choice of fit is not appropriate and not faithful to the real peaks and troughs in the data.

*376-77. This statement does not accurately reflect the results. Note something like a 10-fold increase in MAP from hyper-arid to arid to semi-arid is accompanied by <2-fold increase in median erosion rates. Why is that? Across this range, ground surfaces typically go from being totally bare to having complete seasonal vegetation cover.*

Discussion has been rewritten.

*388- This statement needs some rethinking. I do not follow why erosion 'rates might not be expected to change much' after glacial retreat. What is that based on? I expect rather the opposite as bare sediment and bedrock surfaces are colonised by vegetation over the postglacial period and large areas in North America and Scandinavia are uplifted isostatically. The paraglacial regime is associated with a well studied trajectory of sediment flux over the postglacial period.*

We have rewritten the Discussion and we have also acknowledged biases and uncertainties in the data in formerly glaciated regions.

*389-90. This also ignores that high erosion rates are integrated over short periods...*

We have added the explanation of why shorter timescale may still reflect the influence of former glacial and periglacial processes indirectly.

*397-98. Please clarify.*

We suggest that in mid- and high-latitude regions, glacial erosion increases erosion rates during colder periods whilst in warmer periods, the erosion rates are lower (similar to the rates of tropical regions) because of wider vegetation cover.

*413-414. But according to Fig. 1, it seems that few data are available for boreal regions. The regions for which data are available are some of the most agriculturally exploited on Earth. Could the low SS loads be the result of generally low relief and the intensively modified lowland riverbank revetment?*

Fig. 3 below shows the locations of short-term erosion rates in the Cold K–G climate zone. Although several data are located in central North America with intensive agricultural activity, many are in Canada, Iceland, Scandinavia, and northern and eastern Siberia, with limited anthropogenic influences.

[Figure]

**Figure 3: Locations of short-term erosion rate data in the Cold K–G climate zone.**

*429-36. This is a reasonable deduction, but not a satisfactory resolution. Why not recompute the data using a range of different thresholds to evaluate the problem thoroughly? As for the second point, I don't follow the logic. How does this explain the discrepancy? You find an 8 vs 3.5-fold increase in the smaller basins but then point to the conservation efforts in upland (smaller) basins?*

Thanks for the suggestion. However, due to the limited resolution of the data source and data availability between various percentages of anthropogenic influences, it is not possible to try different thresholds. The 3.5 and 8-fold differences between anthropogenic and non-anthropogenic regions in large and small basins, respectively, were reported by Dedkov and Mozzherin, (1996), not by our study.

*438-41. Yes, slope forms part of stream power law, but slope was not part of your evaluation of the influence of drainage area. It's difficult to see the point of this statement.*

We have removed the statement of slope to avoid confusion.

*442- Be10 derived erosion rates are rarely affected by floodplain storage—the sediment would need to be buried deeply for a long time (>$10^5$ y), then large volumes would need to be somehow reworked into the channel. As for '...violating the detachment limited assumption within area-erosion relationships', it's hard to follow what this is getting at—certainly not the effects of sediment storage on Be10 erosion rates. Further, the Whipple et al. (1999) reference cited here makes no mention of cosmo.*

We have removed 'cosmogenic radionuclide-derived erosion rates' since our result shows unclear relationships between basin area and both long- and short-term erosion rates. We also clarified that the influence of floodplain storage is more critical for the short-term erosion rates.

*438-52. A paragraph of confused thinking. E.g. 442-446, this is not an argument that accounts for differential erosion rates in large vs small basins. 447, What evidence is there that active plate margins have steeper relief than passive margins? The steepness of hillslopes is set essentially by rock strength such that mass failure*

*occurs beyond a certain threshold of internal friction within the slope (Schmidt & Montgomery 1995, Science). Elsewhere in the MS, 126, it is stated that tectonic uplift lowers rock strength via increased fracturing (also not true as a rule: the tallest hillslopes commonly occur in tectonically active terrain, e.g., Nanga Parbat).*

The entire Discussion has been rewritten and all this has been removed.

*449- It is clearly true that lithology strongly influences erosion rates, but this analysis did not assess lithology. Why is this being raised here?*

The purpose of this statement is to say that lithology, as a potential control on erosion rate, but which is not related to basin size, may partly be causing the unclear relationship between basin size and erosion rate.

*461- It is not clear whether this statement refers to short-term or long-term erosion rates but in the case of Be10 derived rates the sediment buffering is not very effective. Several studies by Wittman have shown this.*

This statement refers to short-term erosion rates. Short-term variations of erosion rates are harder to detect at the outlet of large basins due to the higher buffering capacity.

*466-73. One of the most striking aspects of Fig. 7 is that tropical basins have much higher $R_{S/L}$ combined with a greater sensitivity to drainage area. I would expect to find some Discussion of that point but I can find no explanation for why $R_{S/L}$ ratios are notably higher, nor why such regions are more sensitive to drainage area.*

Discussion has been rewritten and includes an explanation of this Fig (lines 515-530).

*489- As noted above, the representation of local channel gradient in Fig. 8 conflates the Be10 derived basin-scale erosion rates with reach-scale fluvial incision rate. Be10 abundances measured in fluvial sediment are not closely related to basin-scale erosion rates.*

The mean channel gradient was calculated from the entire channel, not from the reach-scale (local) gradient, as explained above.

*492-499. Clearly true; agriculture is highly concentrated in lowland settings. But its effect in terms of soil loss is likely to be most destructive in steep terrain. This study sets out to compare erosion rates. And yet, the last few sentences reveal the failure of SS load data to capture soil loss where it actually occurs on hillslopes—due to sediment trapping in reservoirs and perhaps post hoc soil conservation efforts. Recent advances in isotope-based approaches (e.g., cosmogenic C14) mean that soil depletion can be quantified without the source-to-sink assumptions inherent with conventional sediment yield estimates.*

This is an interesting point but beyond the scope of the paper.

**References (beyond those in the manuscript):**

Alexandrov, Y. et al. (2009) Suspended sediment load, bedload, and dissolved load yields from a semi-arid drainage basin: a 15-year study. Water Resources Research, 45, doi: 10.1029/2008WR007314

Cleveland, W. S. (1979) Robust locally weighted regression and smoothing scatterplots, Journal of the American Statistical Association, 74, 829–836, https://doi.org/10.1080/01621459.1979.10481038.

Sadler, P. M. (1981) Sediment accumulation rates and the completeness of stratigraphic sections, The Journal of Geology, 89, 569–584, https://doi.org/10.1086/628623.

Sadler, P. M., and Jerolmack, D. J. (2014) Scaling laws for aggradation, denudation and progradation rates: the case for time-scale invariance at sediment sources and sinks, Geological Society, London, Special Publications, 404, 69–88, https://doi.org/10.1144/SP404.7.

Thornes, J.B. (Ed) (1990) Vegetation and Erosion, Wiley.

---

## Referee Report (RR1)

Review to ESurf-2021-7 by Chen et al (R2)

I am reviewing this paper now for the second time, so I am familiar with it. The general idea is that a global comparison between decadal-scale suspended sediment yields and longer-term cosmogenic nuclide-derived "erosion" rates allows to draw conclusions about what is controlling either. Although the manuscript has benefitted from a general streamlining of the overall message (i.e. now mainly focusing on the influence of climate), I still find it wordy, the discussion needs re-structuring (see below) and the overall suggestions by the reviewers have in my view not been adequately addressed. Below, I focus on 6 major issues that should be dealt with before the paper is ready to be anything close to being published.

1) One issue (still) regards the integration time scale of cosmo rate: The reviewers suggest to go into more depth e.g. regarding the fact that the integration time scale of cosmo rates is a function of the "erosion" rate itself, and also includes weathering (being hence a denudation rate!), in contrast to sediment yield data. As far as I read the text, the cosmo integration time scale is only very generally treated by saying that these rates "integrate over 10^3 to 10^6 years" (l. 234-235), and hence the "data covers several glacial-interglacial cycles". Perhaps the authors are not aware of the fact that by diving by 60 cm (rock) or 100 cm (soil) depth scale, the integration time scale can be calculated from the denudation rate, and this type of information can be used to discuss the data and the trends with e.g. climate proxies. The authors say they have "clarified" that issue, but fail to include this anywhere in the actual discussion of the trend in D vs Precip. The authors ignore a great paper by Schaller & Ehlers, 2006, that looked at the "limits of quantifying climate driven changes in denudation rates with cosmogenic radionuclides"- although the paper is more focused on denudation records through time, a lesson the authors could learn from it that cosmo rates have this inherent variable integration time scale, which should affect the pattern with recent (!) MAP they observe. In my view, they should first show that climate hasn't changed for the regions they use data from, and then they can start making the argument that cosmo rates and MAP follow some trend. Just simply saying that they use recent MAP data because everyone else does that seems a bit too oversimplified. There are global climate models that could be inspected for that purpose. One way of dealing with the integration time scale of cosmo rates would be to include a smoothing function in Fig. 3 (i.e. the "error bar" of the smoothing function could be linked to the integration time scale of the y-axis, the denudation rate). Where higher rates occur, integration times scales are shorter, and longer for lower rates. How does that affect the trend shown in Fig 3a?

2) Issue 2 is on the Sadler effect. Note a paper by Wilkinson 2005 (The Journal of Geology) that uses precipitation amounts and duration as analogue for the effect of hiatuses on sedimentation rates. It is generally describing the effect of hiatuses on datasets, such as any rate determined over some length of time (e.g sediment accumulation rates, erosion rates, bedrock incision rates, but also precipitation rates etc..). Hence, it would be necessary in such a global comparison (that does explicitly NOT compare 1:1 sites of where both cosmo and sediment yield have been measured *at the same location*) to test whether the sediment yield data and/or

precipitation rates are biased by the Sadler effect (e.g. by plotting the rate versus measurement interval). I assume that not all sediment yield data nor precipitation date have the same measurement interval ?! As one reviewer puts it: "Whether or not the SS load data reported here is subject to a time-dependent bias is for the authors to demonstrate."

3) I guess the most important issue regards vegetation cover. The authors motivate their findings (cosmo trend with MAP) with the early findings of e.g. Langbein & Schumm, where peaking erosion rates fall together with a transition from dry to wet precipitation and sparse to extensive vegetation. Although this interpretation is generally fine with me, I am wondering why the effect of recent vegetation (not land use) was not inspected in more detail, by using recent global vegetation maps?. Using a LGM vegetation map the authors explain the higher cosmo rates in the cold zones by glacial and periglacial influence. In my view, this finding is nothing new and does not need to be discussed in detail. I would rather have expected a deeper analysis of vegetation versus precipitation effects in the other climate zones. A recent paper by Starke et al (2020) in Science showed that vegetation and precipitation co-vary and interact, with different feedback strengths. Therefore it would be highly interesting to include vegetation datasets into the analysis.

4) Overall, I find the organization of the discussion still wholesale confusing, and still not much to the point. They set out with their "key finding" (l. 389) regarding the mentioned trend in cosmo rates with MAP. Without relying (again) on the interpretation made by others, I would suggest to include the vegetation dataset and try to develop the discussion from that- mainly because I think that the discussion points that follow do not help (as organized in this order, they need to be re-arranged!) and only add confusion. At this point (i.e. after mentioning the k"ey finding"), I would have expected a discussion on why cosmo rates may follow this trend (Fig 3a). Perhaps the authors are saying that some of this trend is controlled by glacial/periglacial processes as suggested what they write in lines 433 ff? (motivated by the long averaging time scale of cosmo rates?) But how would that affect the trend shown in Fig 3a? Are they saying that the peak around 500-700 mm MAP is an artifact of glacial processes inherited because of the long integration time scale? I guess the fact that this discussion is interrupted by talking about the absence of this trend for the short-term rates (lines 402-416) does not help either. In general, it does not help that the authors jump around in the discussion between the trend shown in Fig. 3a, and the observations drawn from comparisons between long- and short-term rates in the other figures.
I guess a summary suggestion on this trend shown in Fig 3a is: The authors need to more thoroughly inspect possible mechanisms, including: effects of integration time, effects of changes in MAP, vegetation cover, and lastly, possible bias within cosmogenic nuclide analysis itself (yes, sorry). The latter must include some discussion on weathering underestimated from cosmo data because of thick weathering profiles in the tropics (e.g. Riebe et al., 2001, Dixon et al., 2009). In tropical basins, denudation rates might be underestimated as mineral dissolution at depth is not "seen" by cosmo. Naturally, this effect could help to explain why short-term rates are so much higher in the tropical climate zone than cosmo rates. So, simply acknowledging that there are biases in cosmo rates does not help, either-they need to be explicitly mentioned and discussed.

5) I think I raised this concern in my first review that a 1.4-fold increase in short-term erosion rates when comparing anthropogenically and non-anthropogenically impacted regions is in my view not a "significant" increase (Fig. 5). Especially in the light of what is written in lines 402-416 that make it sound like the USA dataset, and also the entire short-term dataset because of this high variability), may not be representative enough to draw such a conclusion.

6) I think the last two paras of the discussion would benefit from being clearly separated from the rest of the discussion. But that would not solve a profound issue that some of these controls cannot be separated by other effects. A way of dealing with different components that might interact with each other would be to do a principle component or factor analysis, that might reveal which component is more influential. I question hence the separate treatment of these factors such as channel gradient, catchment area, in comparison to vegetation, climate and so on. Perhaps I should have mentioned this as first point in my list, so, mentioning this at last does not mean it is the least significant.

---

## Editor Decision (ED1)

[revised manuscript text omitted]

---

## Author Response (AR2)

Dear Prof Mudd,

Thank you for providing us with further detailed and helpful reviews of our manuscript. We fully agree and acknowledge that this manuscript is attempting something different and that the analysis contains many uncertainties arising from the data sources and their assumptions. However, we aim for this paper to be a stepping stone in a wider conversation about the links between climate and erosion – not a definitive answer. We hope it sparks a debate in the community and provides some new perspective. We have therefore, attempted to make all the uncertainties and caveats as clear as possible and we have restructured the Introduction (and added subsections) and the Discussion. However, please do let us know if we need to clarify anything further. Below, we summarise briefly the requested revisions and the corresponding lines in the manuscript in response to each. There are several requests from the Reviewer which we feel are out of scope with our paper and we also justify this below. We hope the new revisions add clarity to our paper.

Best wishes,

Shiuan-An Chen, Katerina Michaelides, David Richards, and Michael Singer
* * *
In a nutshell the requested additions that we address in the manuscript are:

**1) Acknowledge and explain Sadler effect.**

We have added discussion to manuscript (new lines 167–174).

**2) CRN integration timescales. (AE and Reviewer)**

We have added discussion to manuscript (new lines 188–192)

**3) Unpick tectonic from climate signal (AE)**

We have added a new section that analyses the relationship between climate and erosion in tectonic and non-tectonic regions (new lines 352–363, 442–450, and new figures 4, 10b, A1, A2)

There are several suggestions from the reviews that we believe are not in scope with our paper:

**4) Use climate models to "inspect" past climate change (Reviewer)**

This is not a trivial request and the huge effort required to do this would yield very little useful quantitative information that could be used in the paper. Paleoclimate models are full of uncertainties themselves, and while they can indicate where climate was different in the past, we would then not be able to use this information in any quantitative way (e.g. extract meaningful MAP) with the erosion data. I assume the Reviewer would want us to look at climate at different points in the past (e.g. 1000, 5000, 10000, 30000 years ago etc.). This is not as simple or trivial as they imply! There are numerous paleoclimate models, each with their own assumptions, limitations, and uncertainties. We explicitly state our assumption (and its limits) about climate change in the manuscript.

**5) Correct for chemical vs physical weathering (AE)**

While it does make sense to make this correction to allow for a more direct comparison between sediment flux measurements and CRNs, this is not a trivial exercise due to lack of relevant data. If the average offset between erosion rate and denudation rate could be provided, we'd be happy to correct our plots accordingly. Sadly, the OCTOPUS database does not contain the relevant info on erosion rates (correcting for chemical weathering), and we have no intention in going back to the thousands of entries in this database to compute the chemical weathering component, especially because the relevant info would not be available for most studies. More to the point, this step is not necessary to broadly compare denudation rates across different landscapes, as chemical weathering is undoubtedly small compared to physical erosion. It is also not germane to exploring the relative differences between erosion rates across different climate categories, which is what we have done here. Furthermore, we don't have co-located sediment flux and CRN data, so using sediment flux (and associated dissolved load) is not viable. We have added a sentence or two explain how climatic variations in chemical weather might be (or not) reflected in the CRN data.

**6) Use global vegetation map to investigate role of vegetation on erosion (Reviewer).**

The focus of our paper is on the role of climate. We discuss vegetation in this story as a likely factor that co-varies with climate, but it is beyond the scope of the paper to include a whole separate analysis of global vegetation as suggested by the reviewer. This would be something to follow up with potentially. Moreover, the Köppen-Geiger climate index includes vegetation within it.

**7) Do principle [sic] component analysis for all factors (Reviewer).**

We do not feel such an analysis would add much to this paper. In fact, we feel it would simply add more confusion to an already complicated story. Many authors infer too much explanatory power to PCA, but it is only a statistical method and does not clarify causal mechanisms. Anything further would lengthen (not shorten) the paper and add confusion, given the large number of variables included.

Other reviewer comments:

- *I think I raised this concern in my first review that a 1.4-fold increase in short-term erosion rates when comparing anthropogenically and non-anthropogenically impacted regions is in my view not a "significant" increase (Fig. 5). Especially in the light of what is written in lines 402-416 that make it sound like the USA dataset, and also the entire short-term dataset because of this high variability, may not be representative enough to draw such a conclusion.*

As we explained to the reviewer's comment before, the magnitude of a difference between medians does not indicate whether or not the difference is significant. Significance in this case is tested via the Kruskal–Wallis test, which showed that these two distributions are indeed statistically different (see manuscript). However, we also discussed the potential reasons why the difference is smaller than was shown in the literature.

---

## Author Response (AR3)

Dear Prof Mudd,

Thank you for your insightful comments and suggestions on the latest version of our manuscript. Below we address all your comments. In addition, we have carried out some extra analyses to make the paper more consistent in the comparison between long- and short-term erosion rates. Specifcally, we plotted all the global long- and short-term data against MAP (derived from the GPCC gridded dataset) – previously we had just plotted these for the USA. We further plotted median erosion rates from each basin (both long- and short-term) against MAP to check for any biases in spatial sampling (no significant biases found). We fitted a LOWESS regression through the short-term vs MAP relationship (as for the long-term) and found a similar pattern (albeit with higher overall rates). We have edited the manuscript to reflect these changes. We hope these corrections are to your satisfaction – but please do get back to us if there are any further issues or queries.

Best wishes,

Shiuan-An Chen, Katerina Michaelides, David Richards, and Michael Singer
* * *
**1. Co-located short- and long-term erosion rate data.**

We had originally done this analysis but did not include it because the *n* of co-located points was so small (*n* = 79). However, we have now added a new figure with the co-located points (new Fig. 3) which shows a similar pattern as the global dataset (Fig. 2b) which combines all data points within each climate zone.

**2. The data of Figures 3, 4, and 10 are from the USA only.**

We originally only analysed the relationship between MAP and erosion rates in the USA because of the widespread systematic availability of gauge rainfall data across the USA. We have now constructed this scatterplot between long- and short-term erosion rates and MAP globally based on the GPCC rainfall dataset. So the new Figs 4, 5 and 11 (the old 3, 4, and 10) now present global relationships. We have accordingly updated the LOWESS fits for both long- and short-term data and globally, there is now a similar pattern between the two, albeit the short-term erosion rates are typically higher than the long-term rates.

**3. Spatial bias of long-term erosion rate sampling.**

We have now computed the median long- and short-term erosion rates for all drainage basins globally and the overall relationship between MAP and erosion rate remains the same (new supplemental Fig. A1). Therefore, the sampling bias does not translate into any significant influence on the shape of the relationship.

*Line 71: high uplift rates do not have to result in threshold slopes. They could just result in higher relief. I would add "increased relief" before "threshold slopes".*

Done (Line 69)

*Line 73: Tectonics itself doesn't lead to rapid production of sediment during rainstorms. It increases gradients, and the steeper gradients are the cause of the increase sediment production (or actually sediment transport…increased sediment production is because the increased sediment transport results in thinner soils, which leads to increased sediment production). Anyway, this sentence glosses over a lot and could be edited to add some depth.*

Done (Line 72)

*Line 94: typo: "erosion via"*

Done (Line 93)

*Line 141: It is not the erosion rate that determines the CRN concentration, but rather the denudation, or total mass loss rate. Which includes chemical weathering. For accuracy you should say this and then refer back to the point from the previous paragraph that dissolved load is usually quite small. You do say this on line 176 so why not here?*

Done (Line 141)

*Line 151: Say something like: "10Be erosion rates average erosion rates over a characteristic timescale determined by the nuclide concentration divided by an average nuclide production rate. This equates roughly to the time it takes to erode approximately 60 cm of material (Kirchner et al 2001). Therefore a rock lowering rate of 1 mm per year equates to a 600 year timescale, 0.1 mm/yr 6000 years, and so on."*

Done (Line 148)

*Line 265: Typo: "summed"*

Done (Line 254)

*Figure 3: Why are only the USA data used?*

See point 2 above

*Figure 4: Why only the USA?*

See point 2 above

*Figure 5: There us a light purple and a dark purple in the figure but not in the legend. Add both purples to the legend.*

Done (new Fig. 6)

*Line 403: I'm working from a paper copy so can't search, but is R_S/L only defined previously in a figure caption? It has been a while since it was mentioned so I might reiterate what it means here.*

First mentioned in Line 301 (and also in the caption of Figure 2), but we have added in Line 426, too.

*Line 440: I don't think you have really addressed the sampling bias in the CRN data. I am quite confident that if you took a bunch of CRN samples from the Sahara you would get a very low erosion rate. In any climate region you could find a low relief landscape to sample if you wanted to pull the erosion rates down, or a high relief landscape to pull the erosion rates up. Also is this not from the USA data only? That is what is suggested in Figure 3.*

See point 3 above

*Line 443: Again, sampling bias. How do you know it is not driving this trend? This statement appears to be making a global comparison, but figure 3 says the data is from the USA only. If it is from the USA, is the trend because most of the very high erosion rates are from the mountains around Los Angeles, which are heavily sampled and uplifting the fastest of any sampled region in the USA? Also, the large numbers of lower points around 1000-1250 mm/yr: are these not the data from the Appalachians collected by Paul Bierman's students? How were the data aerially weighted? If someone collected large numbers of data from small basins (e.g., in the San Gabriel Mountains) vs a handful from large basins e.g., the handful of sites in Texas, which cover basins that are cumulatively much larger than the dozens of sites in the Transverse Ranges, do these get a greater weighting in the LOWESS regression?*

See point 3 above

*Line 459: This is an interesting point because the earlier papers had a much more limited dataset than you do. As you have collected a very large suspended sediment dataset, you only see noise when relating the flux rates to climate. So the trends from those earlier papers appear to be from sampling bias. How do you know the climate trend you see in the long-term rates is not from sampling bias?*

See point 3 above and the intro to the rebuttal for changes made to the analysis. The LOWESS regression for the short-term erosion vs MAP is now consistent with the long-term LOWESS.

*Figure 10: Again, is this USA data only or global data?*

See point 2 above

*Line 541: Delete reference to stream power law (stream power law uses local, not basin gradient to determine local erosion rate). Also the sentence doesn't need it.*

Sentence removed

*Line 561: This is, at best, misleading. Basin area only affects erosion rate in the stream power law if everything else is equal. But everything else is not equal: gradients get steeper as drainage area goes down. So under the steady state assumption there is \*\*no\*\* dependence of erosion rates on basin area if you are sampling the same basin at two different places. If you are in a active mountain belt where the basis are expected to be eroding at different rates, you cannot just compare the slopes of two basins, or the gradients of two different basins. You have to combine both of these things: they are inseparable if you assume stream power is driving erosion rates (they are also inseparable for virtually any proposed model of channel incision). This is the reason why people use the channel steepness index rather than either gradient or drainage area.*

This sentence about stream power has now been removed.

---

## Author Response (AR4)

Dear Prof Mudd,

Thank you for your suggestions on the latest version of our manuscript. We made edits to all the suggestions in track changes and in the clean version. We hope these corrections can enhance the clarity of this manuscript.

Best wishes,

Shiuan-An Chen, Katerina Michaelides, David Richards, and Michael Singer